# SUMO peptidase ULP-4 regulates mitochondrial UPR-mediated innate immunity and lifespan extension

Kaiyu Gao[1], Yi Li[1,2], Shumei Hu[2], Ying Liu[1]*

[1]State Key Laboratory of Membrane Biology, Institute of Molecular Medicine, Beijing Key Laboratory of Cardiometabolic Molecular Medicine, and Peking-Tsinghua Center for Life Sciences, Peking University, Beijing, China; [2]Academy for Advanced Interdisciplinary Studies, Peking University, Beijing, China

**Abstract** Animals respond to mitochondrial stress with the induction of mitochondrial unfolded protein response (UPR[mt]). A cascade of events occur upon UPR[mt] activation, ultimately triggering a transcriptional response governed by two transcription factors: DVE-1 and ATFS-1. Here we identify SUMO-specific peptidase ULP-4 as a positive regulator of *C. elegans* UPR[mt] to control SUMOylation status of DVE-1 and ATFS-1. SUMOylation affects these two axes in the transcriptional program of UPR[mt] with distinct mechanisms: change of DVE-1 subcellular localization vs. change of ATFS-1 stability and activity. Our findings reveal a post-translational modification that promotes immune response and lifespan extension during mitochondrial stress.
DOI: https://doi.org/10.7554/eLife.41792.001

## Introduction

The ability of an organism to cope with an ever-changing and challenging environment lies in its ability to activate stress responses. Failure to appropriately respond to different stresses and maintain cellular and organismal homeostasis could result in multiple diseases including metabolic and neurodegenerative disorders (*Jovaisaite et al., 2014*; *Lee and Ozcan, 2014*; *Wang and Kaufman, 2012*). Animals respond to mitochondrial stress with the induction of mitochondrial unfolded protein response (UPR[mt]), a surveillance program that monitors mitochondrial function and initiates mitochondria-to-nucleus crosstalk to maintain mitochondrial protein-folding homeostasis (*Benedetti et al., 2006*; *Yoneda et al., 2004*) and coordinate the expression of electron transport chain (ETC) components in mitochondrial and nuclear genomes (*Houtkooper et al., 2013*). UPR[mt] also elicits global changes to reprogram metabolism (*Nargund et al., 2015*; *Nargund et al., 2012*), activate immune responses (*Liu et al., 2014*; *Melo and Ruvkun, 2012*; *Pellegrino et al., 2014*) and extend lifespan (*Durieux et al., 2011*; *Merkwirth et al., 2016*; *Tian et al., 2016*).

UPR[mt] signaling ultimately activates a transcriptional response governed by two transcription factors: ATFS-1 and DVE-1. ATFS-1 contains an N-terminal mitochondrial targeting sequence and a C-terminal nuclear localization sequence. Under normal condition, ATFS-1 is imported into mitochondria, where it is degraded by mitochondrial protease LON. During mitochondrial stress, mitochondrial import efficiency is impaired, resulting in nuclear accumulation of ATFS-1 (*Nargund et al., 2012*). ATFS-1 controls approximately half of the mitochondrial stress response genes, including those encoding mitochondrial-specific chaperones, proteases and immune response genes (*Nargund et al., 2012*). ATFS-1 also regulates genes involved in metabolic reprogramming, such as those functioning in glycolysis (*Nargund et al., 2015*). Another axis of the UPR[mt] transcriptional program relies on DVE-1, a homeobox transcription factor homologous to human SATB1/SATB2. Upon mitochondrial perturbation, DVE-1 translocates from cytosol to nucleus, binds to the open-up

*For correspondence:
ying.liu@pku.edu.cn

Competing interests: The authors declare that no competing interests exist.

**eLife digest** Most animal cells carry compartments called mitochondria. These tiny powerhouses produce the energy that fuels many life processes, but they also store important compounds and can even cause an infected or defective cell to kill itself. For a cell, keeping its mitochondria healthy is often a matter of life and death: failure to do so is linked with aging, cancer or diseases such as Alzheimer's.

The cell uses a surveillance program called the mitochondrial unfolded protein response to assess the health of its mitochondria. If something is amiss, the cell activates specific mechanisms to fix the problem, which involves turning on specific genes in its genome.

A protein named ULP-4, which is found in the worm *Caenorhabditis elegans* but also in humans, participates in this process. This enzyme cuts off chemical 'tags' known as SUMO from proteins. Adding and removing these labels changes the place and role of a protein in the cell. However, it was still unclear how ULP-4 played a role in the mitochondrial unfolded protein response.

Here, Gao et al. show that when mitochondria are in distress, ULP-4 removes SUMO from DVE-1 and ATFS-1, two proteins that control separate arms of the mitochondrial unfolded protein response. Without SUMO tags, DVE-1 can relocate to the area in the cell where it can turn on genes that protect and repair mitochondria; meanwhile SUMO-free ATFS-1 becomes more stable and can start acting on the genome. Finally, the experiments show that removing SUMO on DVE-1 and ATFS-1 is essential to keep the worms healthy and with a long lifespan under mitochondrial stress.

The experiments by Gao et al. show that the mitochondrial unfolded protein response relies, at least in part, on SUMO tags. This knowledge opens new avenues of research, and could help fight diseases that emerge when mitochondria fail.

DOI: https://doi.org/10.7554/eLife.41792.002

chromatins devoid of H3K9me2, and initiates the transcription of mitochondrial stress response genes (*Haynes et al., 2007*; *Tian et al., 2016*). While several core components of UPR$^{mt}$ have been identified, the regulation, especially post-translational regulation of these components has not been reported.

The Small Ubiquitin-like Modifier (SUMO) post-translational modifies a large number of proteins that function in diverse biological processes, including transcription, chromatin remodeling, DNA repair and mitochondrial dynamics (*Gill, 2004*; *Hay, 2005*; *Hendriks et al., 2014*; *Prudent et al., 2015*; *Wasiak et al., 2007*; *Yeh et al., 2000*). Growing evidence suggests that rather than modifying a single protein, SUMO often targets multiple proteins within a complex, or within a pathway (*Chymkowitch et al., 2015*; *Hendriks et al., 2014*). Similar to ubiquitination, conjugation of SUMO to its substrates involves an enzymatic cascade including an E1 activating enzyme, an E2 conjugating enzyme and E3 ligases that determine the specificity (*Flotho and Melchior, 2013*). SUMOylation is also a dynamic process, which can be reversed by a family of conserved Sentrin/SUMO-specific proteases (SENPs) (*Mukhopadhyay and Dasso, 2007*). In *C. elegans*, the SENP family consists of four SUMO proteases (ubiquitin-like proteases, ULPs) ULP-1, ULP-2, ULP-4 and ULP-5. Among them, ULP-2 has been reported to deSUMOylate E-cadherin and promotes its recruitment to adherens junctions (*Tsur et al., 2015*). Moreover, ULP-4 has been reported to deSUMOylate HMGS-1 to control mevalonate pathway activity during aging (*Sapir et al., 2014*).

Aberrant activity of SUMOylation drastically affects cellular homeostasis and has been linked with many diseases (*Flotho and Melchior, 2013*; *Mo et al., 2005*; *Sarge and Park-Sarge, 2009*; *Seeler et al., 2007*). It has been reported that SUMO could covalently modify Drp1, a protein essential for mitochondrial dynamics (*Prudent et al., 2015*). In addition, SUMOylation of a pathogenic fragment of Huntingtin, a PolyQ-repeats protein that specifically binds to the outer membrane of mitochondria and impairs mitochondrial function, has been reported to exacerbate neurodegeneration in a *Drosophila* Huntington's disease model (*Costa and Scorrano, 2012*; *Panov et al., 2002*; *Steffan et al., 2004*).

In the present study, we find that under mitochondrial stress, SUMO-specific peptidase ULP-4 is required to deSUMOylate DVE-1 and ATFS-1 to activate UPR$^{mt}$ in *C. elegans*. ULP-4 is also required

to promote UPR^mt-mediated innate immunity and lifespan extension. Our results reveal an essential and unexplored function of post-translational regulation in UPR^mt signaling.

## Results

### SUMO-specific peptidase ULP-4 is required for the activation of UPR^mt

Previously, we have performed a genome-wide RNAi screen to identify genes that are required for the activation of mitochondrial unfolded protein response (UPR^mt) in *C. elegans* (*Liu et al., 2014*). *ulp-4*, a gene encoding ortholog of SUMO-specific peptidase in *C. elegans*, is one of the hits from our primary screen. RNAi of *ulp-4* impaired the activation of UPR^mt that is induced by mitochondrial inhibitor antimycin A, or RNAi of nuclear encoded mitochondrial gene *spg-7* (mitochondrial metallo-protease) (*Figure 1A–B* and *Figure 1—figure supplement 1A*). RNAi of *cco-1* (nuclear-encoded cytochrome c oxidase-1 subunit) is also widely used to disrupt mitochondrial function and activate UPR^mt (*Durieux et al., 2011*; *Nargund et al., 2012*; *Pellegrino et al., 2014*). Consistently, deficiency of *ulp-4* also suppressed the induction of endogenous mitochondrial chaperone genes *hsp-6* and *hsp-60* under *cco-1* RNAi (*Figure 1C* and *Figure 1—figure supplement 1B*). Notably, transcript level of *ulp-4* was also elevated during mitochondrial stress (*Figure 1—figure supplement 1C*). In contrast, *ulp-4* RNAi did not affect the induction of endoplasmic reticulum (ER) stress reporter *hsp-4p::gfp* nor heat shock stress reporter *hsp-16.2p::gfp* (*Figure 1D–E*). In addition, worms treated with *ulp-4* RNAi were still able to induce the expression of endogenous *hsp-4* or *hsp-16.2* during ER or heat shock stress (*Figure 1F*). RNAi of *ulp-4* from L1 stage only delayed worm development a bit (*Figure 1—figure supplement 1D*). To further exclude the possibility that the suppression of UPR^mt by *ulp-4* RNAi is due to developmental delay, we also treated worms with *ulp-4* RNAi starting at L4 stage and observed the reduction of UPR^mt as well (*Figure 1—figure supplement 1E*). Overexpression of ULP-4 in *ulp-4* RNAi worms rescued UPR^mt activation (*Figure 1G* and *Figure 1—figure supplement 1F*). Lastly, we crossed an *ulp-4(tm1597)* mutant allele that lacks 404nt in the promoter region of *ulp-4* with *hsp-6p::gfp* reporter, and showed that the induction of UPR^mt was also impaired in *ulp-4* mutants (*Figure 1H–I* and *Figure 1—figure supplement 1G*).

To see if other SUMO peptidases have similar effects to regulate UPR^mt, we treated *C. elegans* with *ulp-1*, *2*, or *5* RNAi (*Figure 1J*) and tested for their abilities to induce UPR^mt. Deficiency of *ulp-1*, *2*, or *5* failed to suppress antimycin- or *spg-7* RNAi-induced UPR^mt (*Figure 1K*), suggesting a specific role of *ulp-4* in mediating UPR^mt. Sequence alignment (*Katoh and Standley, 2013*) of *ulp-1*, *2*, *4* and *5* revealed that the only conserved region among them is the catalytic domain (*Figure 1—figure supplement 2*). Thus, the specificity of ULP-4 in UPR^mt signaling might be due to its ability to specifically interact with other protein components in UPR^mt pathway.

Conversely, RNAi of the E1 SUMO activating enzyme *aos-1* or the E2 SUMO conjugating enzyme *ubc-9* in *C. elegans* induced UPR^mt more potently (*Figure 1—figure supplement 3A–C*). Moreover, RNAi of *smo-1*, the only SUMO ortholog gene in *C. elegans*, induced only weak UPR^mt under unstressed condition (*Figure 1—figure supplement 3D–E*). However, upon mitochondrial stress, *smo-1* RNAi further activated UPR^mt (*Figure 1—figure supplement 3F–G*). More importantly, *smo-1* RNAi rescued *ulp-4* deficiency-suppressed UPR^mt (*Figure 1—figure supplement 3F–G*). Taken together, these results suggest that ULP-4 plays a specific role to mediate mitochondrial stress response through its SUMO peptidase activity.

### ULP-4 deSUMOylates DVE-1 at K327 residue during mitochondrial stress

To understand the molecular mechanism of ULP-4 in mediating UPR^mt, we sought to identify its protein targets. We first performed a cherry-picked yeast two-hybrid screen to test if ULP-4 could interact with known UPR^mt pathway components. We found that DVE-1, a homeodomain-containing transcription factor in UPR^mt (*Haynes et al., 2007*), interacted with ULP-4 (*Figure 2A*). Notably, DVE-1 could specifically interact with ULP-4, but not ULP-2 or ULP-5 (Note: overexpression of ULP-1 in yeast is lethal) (*Figure 2—figure supplement 1A*). Consistently, *smo-1* is one of the top hits from our yeast two-hybrid screen with DVE-1 as bait (*Figure 2—figure supplement 1B*).

Four possible mechanisms may explain the interaction between SMO-1 and a prey protein in the yeast two-hybrid experiment (*Figure 2B*): (I) SMO-1 covalently modifies the prey; (II) SMO-1 modifies

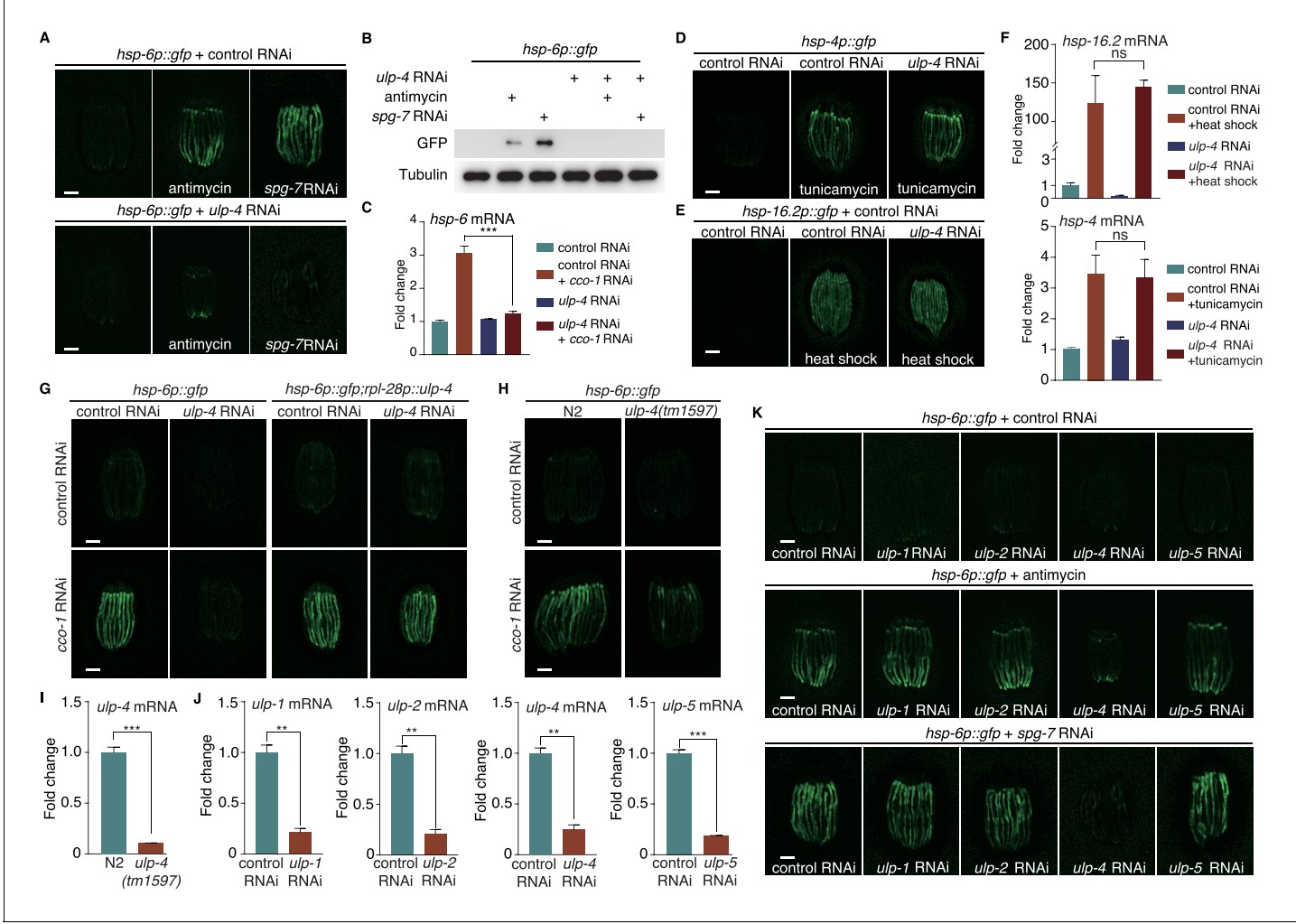

**Figure 1.** SUMO-specific peptidase ULP-4 is required for the activation of UPR[mt]. (**A**) *hsp-6p::gfp* animals fed with control RNAi (top) or *ulp-4* RNAi (bottom) were untreated, or treated with antimycin or *spg-7* RNAi. Scale bar is 200 μm in this study unless otherwise indicated. Secondary RNAi was added 24 hr later after first RNAi unless otherwise indicated. Antimycin was added 48 hr later after first RNAi unless otherwise indicated. (**B**) Immunoblot of GFP expression in untreated *hsp-6p::gfp* worms, or *hsp-6p::gfp* worms treated with antimycin or *spg-7* RNAi. (**C**) Quantitative PCR of endogenous *hsp-6* mRNA levels. (**D–E**) *hsp-4p::gfp* (**D**) or *hsp-16.2p::gfp* (**E**) animals on control or *ulp-4* RNAi were treated with tunicamycin (**D**) or heat-shocked (**E**). (**F**) Quantitative PCR of endogenous *hsp-16.2* and *hsp-4* mRNA levels. (**G**) *hsp-6p::gfp* or *hsp-6p::gfp; rpl-28p::ulp-4* (*ulp-4* overexpression, OE) animals fed with control or *ulp-4* RNAi were untreated or treated with *cco-1* RNAi. Overexpressed *ulp-4* is codon optimized. (**H**) *hsp-6p::gfp* or *ulp-4(tm1597); hsp-6p::gfp* animals were fed with control or *cco-1* RNAi. (**I**) Quantitative PCR of *ulp-4* mRNA level in wild-type or *ulp-4(tm1597)* animals. (**J**) Quantitative PCR of endogenous *ulp-1,2,4,5* mRNA levels under each respective RNAi. (**K**) *hsp-6p::gfp* animals fed with control or *ulp-1,2,4,5* RNAi were treated with antimycin or *spg-7* RNAi. Error bars show standard deviation. Student's t-test, ns not significant, **p<0.002 and ***p<0.0002. All experiments in this paper, if not specifically indicated, have been repeated for at least three times.

DOI: https://doi.org/10.7554/eLife.41792.003

The following figure supplements are available for figure 1:

**Figure supplement 1.** SUMO-specific peptidase ULP-4 is required for the activation of UPR[mt].
DOI: https://doi.org/10.7554/eLife.41792.004
**Figure supplement 2.** Sequence alignment of worm ULP proteins.
DOI: https://doi.org/10.7554/eLife.41792.005
**Figure supplement 3.** Deficiency of *smo-1* or SUMO conjugating enzymes enhances UPR[mt].
DOI: https://doi.org/10.7554/eLife.41792.006

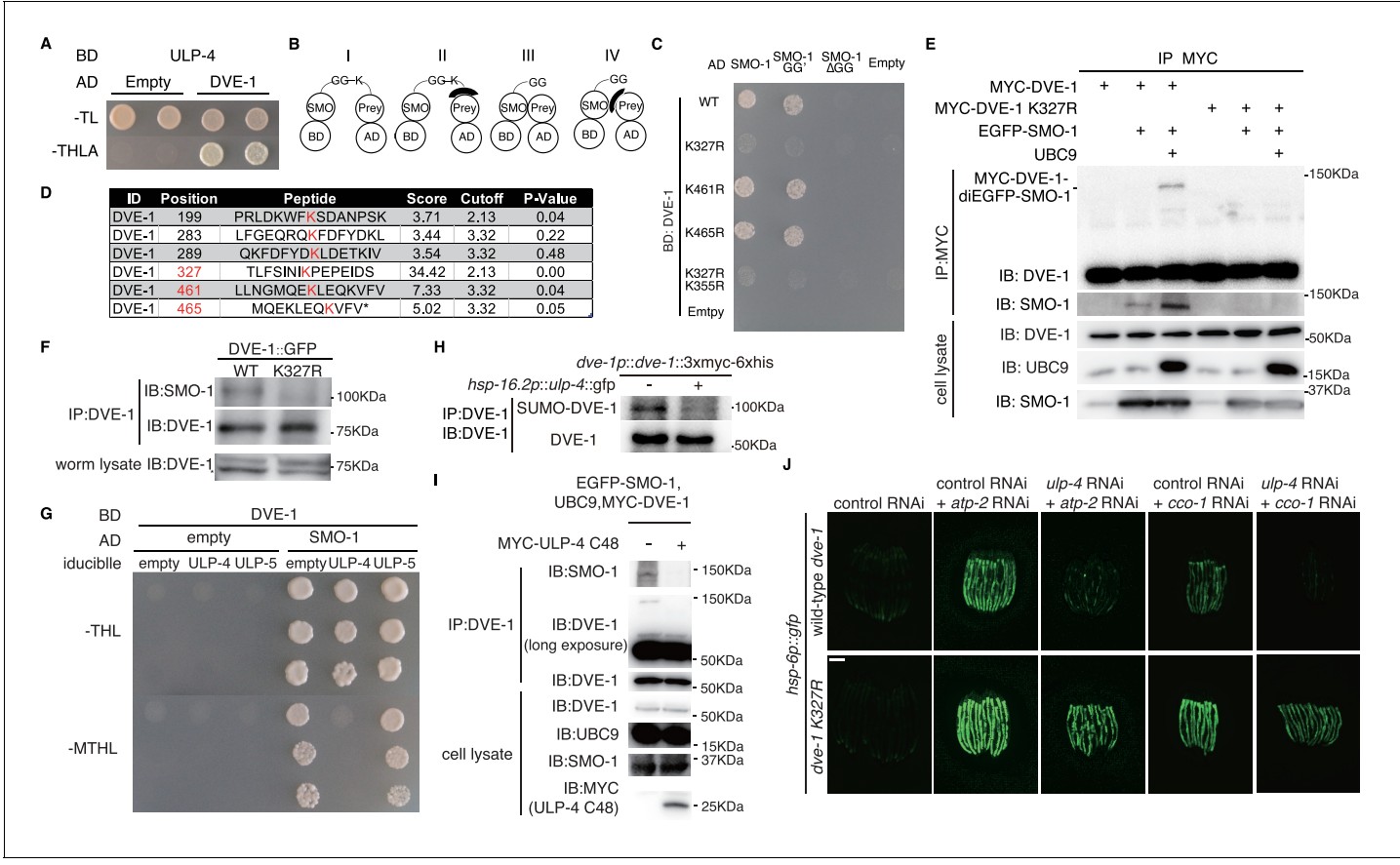

**Figure 2.** ULP-4 deSUMOylates DVE-1 at K327 residue during mitochondrial stress. (**A**) Yeast two-hybrid result of ULP-4 and DVE-1 interaction. AH109 strain was transformed with Gal4-BD-ULP-4 and empty Gal4-AD or Gal4-AD-DVE-1 plasmids. –TL, tryptophan and leucine dropout agar media plates. – THLA, tryptophan, leucine, histidine and adenine dropout agar media plates. (**B**) Diagrams showing possible mechanisms for SMO-1 and prey protein interaction in yeast two-hybrid assay. (**C**) Yeast two-hybrid assay of the interaction between wild-type, K327R, K461R, K465R or K355R DVE-1 and SMO-1. SMO-1 GG', residues after C-terminal di-glycine were deleted. SMO-1ΔGG, di-glycine residues in SMO-1 were deleted. (**D**) Predicted SUMOylation sites of DVE-1 using GPS-SUMO web service (http://sumosp.biocuckoo.org/online.php). (**E**) In vivo SUMOylation assay for DVE-1. Myc-tagged wild-type DVE-1 or DVE-1 K327R, SMO-1 and E2 SUMO conjugating enzyme UBC9 were expressed in 293 T cells. (**F**) DVE-1 SUMOylation in worms. DVE-1::GFP or DVE-1 K327R::GFP driven by *dve-1* promotor were expressed in worms. (**G**) Yeast three-hybrid assay of DVE-1 deSUMOylation. Constitutive Gal4-BD-DVE-1 expression and inducible ULP-4/ULP-5 expression elements were cloned into pBridge. ULP-4/5 was induced when methionine was dropout. – MTHL, methionine, tryptophan, leucine and histidine dropout agar plates. (**H**) Immunoblot of DVE-1 SUMOylation. L4 stage *dve-1p::dve-1::3xmyc-6xhis* worms or *dve-1p::dve-1:: 3xmyc-6xhis; hsp-16.2p::ulp-4::gfp* worms were heat shocked at 37°C for 1 hr, and cultured at 20°C for 8 hr before immunoprecipitation. (**I**) In vivo deSUMOylation assay for DVE-1. Myc-tagged ULP-4 C48 domain, SMO-1 and E2 SUMO conjugating enzyme UBC9 were expressed in 293 T cells. (**J**) *hsp-6p::gfp* or *dve-1* K327R; *hsp-6p::gfp* animals fed with control or *ulp-4* RNAi were treated with control, *atp-2* or *cco-1* RNAi.

DOI: https://doi.org/10.7554/eLife.41792.007

The following figure supplement is available for figure 2:

**Figure supplement 1.** DVE-1 interacts with ULP-4 and SMO-1 in yeast two-hybrid assay.

DOI: https://doi.org/10.7554/eLife.41792.008

an adaptor protein, which interacts with the prey; (III) SMO-1 non-covalently interacts with the prey; (IV) SMO-1 non-covalently interacts with an adaptor protein, which associates with the prey. To identify which mechanism explains our result (*Figure 2A*), we deleted C-terminal tail of SMO-1 to expose a conserved di-glycine motif (GG': active form) or deleted the di-glycine motif to inactivate SMO-1 (ΔGG: inactive form). Only wild type SMO-1 or SMO-1 GG', but not SMO-1 ΔGG, interacted with DVE-1 (*Figure 2C* and *Figure 2—figure supplement 1C*), indicating that SMO-1 either covalently modifies DVE-1, or covalently modifies an adaptor protein that associates with DVE-1.

SUMOylated DVE-1 could be detected in worms, the level of which was elevated under *ulp-4* RNAi (*Figure 2—figure supplement 1D*). We therefore used GPS-SUMO web service, which predicts SUMOyaltion sites of a protein, to identify several lysine residues of DVE-1 that could potentially be SUMOylated (*Figure 2D*) (*Ren et al., 2009*; *Zhao et al., 2014*). To further map the SUMOylation site of DVE-1, we first tested the interaction between SMO-1 and fragments of DVE-1 in the yeast two-hybrid experiment. We found that DVE-1 301–468 associated with SMO-1 (*Figure 2—figure supplement 1E*), suggesting that SUMOylation site may resides in 301–468 amino acids of DVE-1. We then employed site-direct mutagenesis to mutate K327, K461 or K465 residue of DVE-1 to arginine, and tested for its ability to associate with SMO-1. K327R, but not other mutations, abolished SMO-1-DVE-1 interaction (*Figure 2C*). Furthermore, SUMOylation of DVE-1 could be detected when we expressed wild type, but not DVE-1 K327R, with SMO-1 and E2 conjugating enzyme UBC9 in 293 T cells (*Figure 2E*). We also noted that the size shift of SUMOylated DVE-1 was about the molecular weight of two EGFP-SMO-1. To exclude the possibility that DVE-1 has another SUMOylation sites in addition of K327, we expressed a fragment of DVE-1 (295–354), which contains only one lysine residue (K327) in this polypeptide in 293 T cells. We found that the size shift of DVE-1 295–354 is corresponding to di-SMO-1 (*Figure 2—figure supplement 1F*), suggesting that DVE-1 only contains one SUMOylation site. Consistently, SUMOylation of DVE-1 on K327 residue was also observed in *C. elegans* (*Figure 2F*). Taken together, these results suggested that SMO-1 covalently modifies K327 residue of DVE-1.

To test if ULP-4 could directly deSUMOylate DVE-1, we employed a yeast three-hybrid experiment to induce the expression of ULP-4 or ULP-5 in yeasts. The expression of ULP-4 or ULP-5 was driven by a *met17* promoter, which could be induced when growth media is deficient for methionine.Induction of ULP-4, but not ULP-5, prevented yeast growth caused by SMO-1-DVE-1 interaction (*Figure 2G*), suggesting that ULP-4 may removes SUMO moiety from DVE-1. Overexpression of ULP-4 in worms decreased SUMOylation level of DVE-1 (*Figure 2H*). In addition, expression of ULP-4 C48, the catalytic domain of ULP-4 (*Letunic and Bork, 2018*) in 293 T cells could deSUMOylate DVE-1 in mammalian cells (*Figure 2I*). Lastly, if during mitochondrial stress, ULP-4 is indeed required to deSUMOylate DVE-1, mutation of K327 to arginine that prevents DVE-1 SUMOylation would bypass the requirement of ULP-4 for UPR^mt induction. Indeed, we found that a CRISPR/Cas9 knock-in strain with DVE-1 K327R mutation bypassed the requirement of *ulp-4* and was capable to activate UPR^mt under *ulp-4* RNAi (*Figure 2J* and *Figure 2—figure supplement 1G*).

## SUMOylation affects the subcellular localization of DVE-1

We next aimed to understand how SUMOylation affects DVE-1 function. During mitochondrial stress, DVE-1 translocates from cytosol to nucleus (*Haynes et al., 2007*; *Tian et al., 2016*) (*Figure 3A and B*). Inactivation of *ulp-4* by RNAi abolished the nuclear accumulation of DVE-1 in *cco-1* RNAi-treated worms (*Figure 3A–C*). Conversely, overexpression of ulp-4 increased the nuclear accumulation of DVE-1 (*Figure 3D*). More importantly, we found that the induction of UPR^mt correlated well with DVE-1 subcellular localization. When we fed worms with *ulp-4* RNAi for one generation and treated their progeny with *ulp-4* RNAi for twenty-four hours to allow efficient *ulp-4* knockdown, and then fed animals with *cco-1* RNAi, we observed cytosolic accumulation of DVE-1 and suppression of UPR^mt (*Figure 3E*). However, when we treated progeny with a mixture of *ulp-4* and *cco-1* RNAi, which induced mitochondrial stress before *ulp-4* was efficiently knocked down, DVE-1 was still able to translocate to the nucleus and induce UPR^mt (*Figure 3E*).

We also tested the subcellular localization of DVE-1 K327R under *ulp-4* RNAi. Different from wild type proteins, DVE-1 K327R constitutively localized in the nucleus of *C. elegans*, even if *ulp-4* was knocked down by RNAi (*Figure 3F*). Conversely, SUMO-mimetic DVE-1 constitutively localized in the cytosol (*Figure 3G*). Taken together, during mitochondrial stress, ULP-4 deSUMOylates DVE-1 at K327 residue to allow its nuclear accumulation to initiate UPR^mt.

## ULP-4 deSUMOylates ATFS-1 at K326 residue upon mitochondrial stress

Aside from DVE-1, we found that a fragment of ATFS-1 (372–472), another transcription factor in UPR^mt, was also able to interact with ULP-4 (*Figure 4A*, note: expression of full-length ATFS-1 is toxic in yeast). Similar as DVE-1, ATFS-1 could only interact with ULP-4, but not ULP-2 and ULP-5

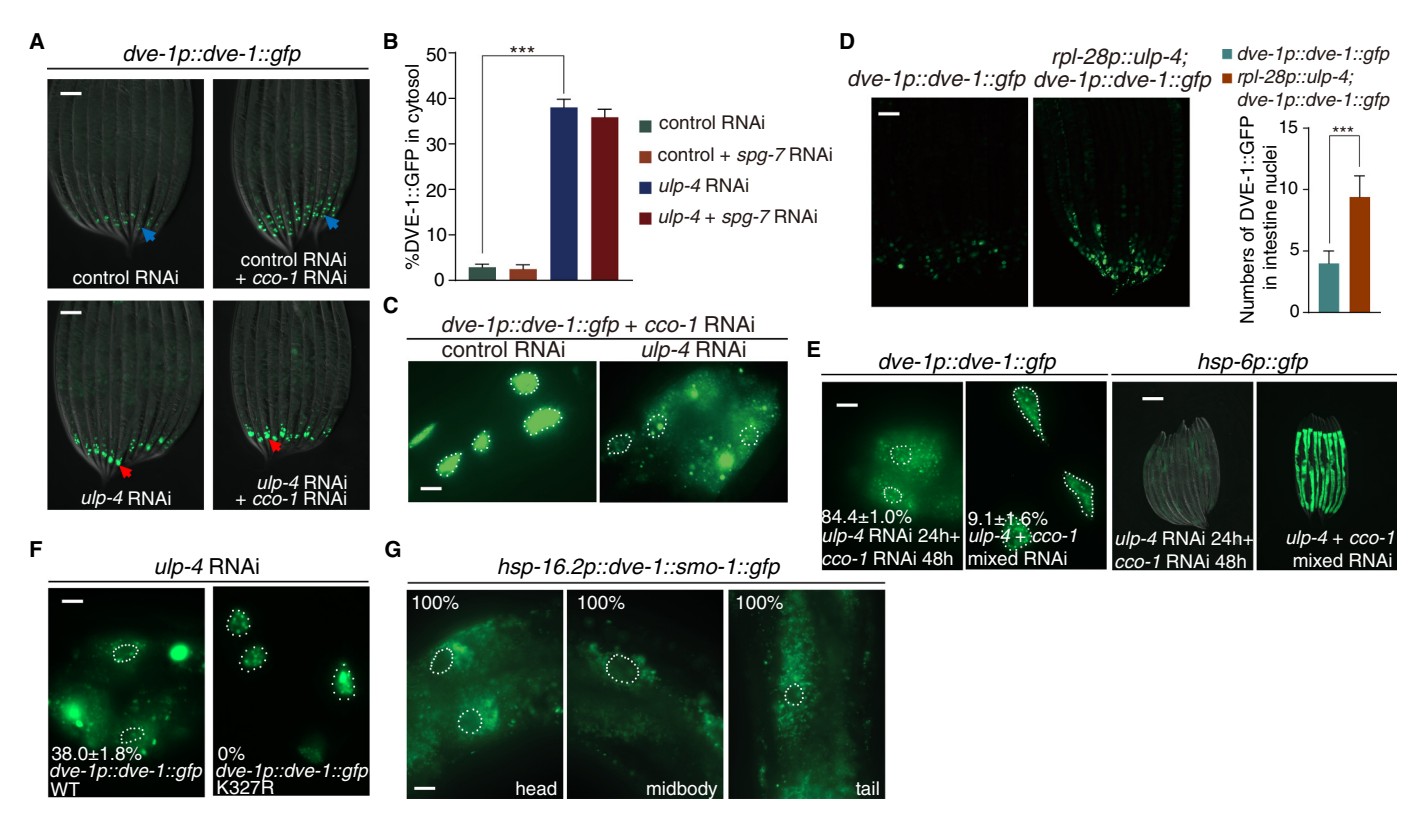

**Figure 3.** SUMOylation affects the subcellular localization of DVE-1. (**A**) Representative fluorescent images of *dve-1p::dve-1::gfp* animals fed with indicated RNAi. Blue arrows indicate that DVE-1::GFP remains in the nucleus. Red arrows indicate that DVE-1::GFP localizes in the cytosol. Scale bar is 100 μm. (**B**) Percentage of DVE-1::GFP in cytosol with indicated RNAi. Student's t-test, ***p<0.0002. Error bars indicate standard deviation. (**C**) Representative fluorescent images of *dve-1p::dve-1::gfp* animals fed with control or *ulp-4* RNAi. Scale bar is 10 μm. (**D**) Representative fluorescent images (left) and statistic result (right) for nuclear accumulation of DVE-1 in *dve-1p::dve-1::gfp* animals with or without *ulp-4* overexpression. (**E**) Induction of UPR[mt] correlates with nuclear accumulation of DVE-1. *dve-1p::dve-1::gfp* and *hsp-6p::gfp* P0 animals were fed with *ulp-4* RNAi. *dve-1p::dve-1::gfp* (left) or *hsp-6p::gfp* (right) animals were cultured with *ulp-4* RNAi for one generation, and F1s were then treated with indicated RNAi starting at L1 stage. Scale bar is 10 μm (left) and 200 μm (right). The number indicates the proportion of animals with DVE-1::GFP in cytosol. Numbers indicate Mean ±standard deviation. N = 3 biological replicates, n > 100 worms each replicate. (**F**) DVE-1 K327R constitutively localizes in the nucleus, even under *ulp-4* RNAi. The number indicates the proportion of animals with DVE-1::GFP in cytosol. Numbers indicate Mean ±standard deviation. N = 3 biological replicates, n > 100 worms per replicate. (**G**) Fusion of SMO-1 to DVE-1 to mimic its SUMOylated form results in DVE-1 expression in the cytosol. Worms were observed 2 hr after 1 hr heat shock at 37°C. The number indicates the proportion of animals with DVE-1::GFP in cytosol.

DOI: https://doi.org/10.7554/eLife.41792.009

(*Figure 4B*). Four lysine residues were predicted by GPS-SUMO tool to be potential SUMOylation sites in ATFS-1 (*Figure 4C*) (*Zhao et al., 2014*). In vivo SUMOylation assay in 293 T cells identified K326 residue as the *bona fide* ATFS-1 SUMOylation site (*Figure 4D*). SUMOylation of ATFS-1 was also observed in *C. elegans*, which could be abolished by mutating K326 residue to arginine (*Figure 4E*). Expression of ULP-4 C48, the catalytic domain of ULP-4, in 293 T cells deSUMOylated ATFS-1 (*Figure 4F*). Lastly, expression of ATFS-1 K326R in *atfs-1(tm4525)* hypomorphic allele bypassed the requirement of *ulp-4* in UPR[mt] activation (*Figure 4G*).

## SUMOylation affects the stability and transcriptional activity of ATFS-1

To see if SUMO also affects ATFS-1 localization, we used *hsp-16.2* heat shock promoter to drive the expression of GFP-tagged full-length ATFS-1. Upon mitochondrial perturbation, ATFS-1 was still able to translocate to the nucleus when ULP-4 expression is diminished (*Figure 5A*). It should be noted that it is very difficult to express full-length ATFS-1 in worms, probably due to toxicity (expression of full-length ATFS-1 in yeast is lethal). Therefore, we made truncations of ATFS-1, and found

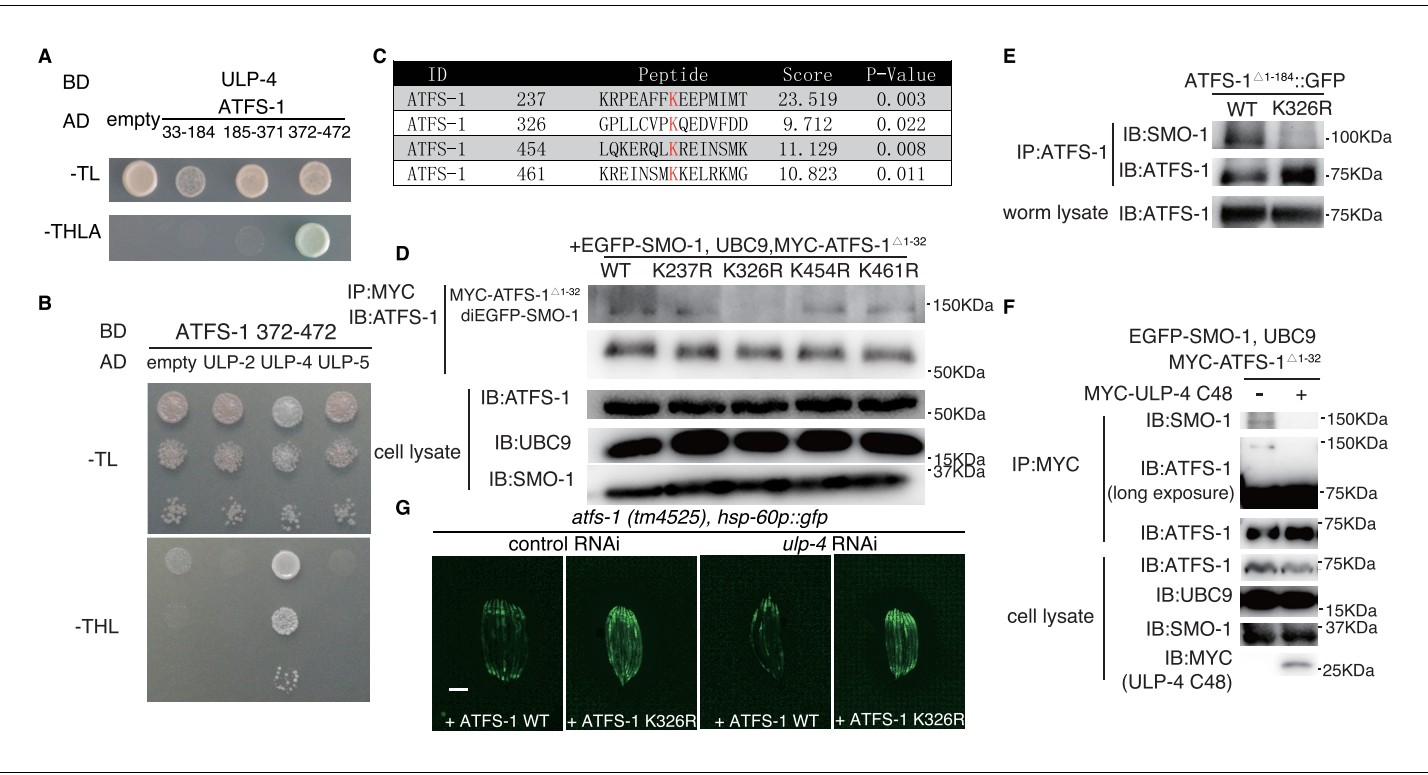

**Figure 4.** ULP-4 deSUMOylates ATFS-1 at K326 residue upon mitochondrial stress. (**A**) Yeast two-hybrid assay for ATFS-1 and ULP-4 interaction. ATFS-1 was truncated into 33–174, 175-371and 372–472 fragments. (**B**) Yeast two-hybrid assay for ATFS-1 372–472 and ULPs. (**C**) Predicted SUMOylation sites of ATFS-1 using GPS-SUMO web service. (**D**) In vivo SUMOylation assay for ATFS-1. Myc-tagged wild-type ATFS-1$^{\Delta 1-32}$ or ATFS-1$^{\Delta\ 1-32}$ lysine mutants, SMO-1 and E2 SUMO conjugating enzyme UBC9 were expressed in 293 T cells. (**E**) ATFS-1 SUMOylation in worms. Wild-type or ATFS-1$^{\Delta\ 1-184}$ K326R was expressed in worms. (**F**) ATFS-1$^{\Delta\ 1-32}$ deSUMOylation in 293 T cells. SMO-1, UBC9, ATFS-1$^{\Delta\ 1-32}$ and ULP-4 C48 domain were expressed in 293 T cells. (**G**) Representative fluorescent images of *atfs-1(tm4525); hsp-60p::gfp* with wild-type ATFS-1 or ATFS-1 K326R animals fed on control or *ulp-4* RNAi.

DOI: https://doi.org/10.7554/eLife.41792.010

that ATFS-1$^{\Delta 1-184}$ expressed well and localized in the nucleus (*Figure 5—figure supplement 1A*). Abolishing SUMOylation site of ATFS-1 (K326R) did not affect subcellular localization of ATFS-1 $^{\Delta\ 1-184}$ either (*Figure 5—figure supplement 1B*).

Interestingly, we noticed that *ulp-4* RNAi greatly reduced the protein level of ATFS-1$^{\Delta\ 1-184}$ (*Figure 5—figure supplement 1C*), suggesting that *ulp-4* RNAi may affect ATFS-1 expression, or stability. Seven hours after heat induction of ATFS-1 expression, levels of ATFS-1$^{\Delta\ 1-184}$ were comparable in control or *ulp-4* RNAi animals (*Figure 5B*). However, after 24 hours, ATFS-1$^{\Delta\ 1-184}$ level in *ulp-4* RNAi treated animals significantly decreased (*Figure 5B*). Treating worms with proteasome inhibitor MG132 partially rescued the reduction of ATFS-1 $^{\Delta\ 1-184}$ under *ulp-4* RNAi (*Figure 5—figure supplement 1C*), suggesting that *ulp-4* RNAi affects ATFS-1 protein stability. Full-length ATFS-1 proteins could be detected when mitochondrial protease LON-1 was inhibited (*Nargund et al., 2012*). Treating worms with *lon-1* RNAi, we showed that full-length ATFS-1 levels were also reduced under *ulp-4* RNAi (*Figure 5C*). Mutation (K326R) that abolished ATFS-1 SUMOylation partially restored its protein level under *ulp-4* RNAi (*Figure 5D–E*). In contrary, fusion of SMO-1 to mimic SUMOylated form of ATFS-1 significantly reduced its protein level (*Figure 5D–E*). Taken together, these results suggest that SUMOylation reduces the stability of ATFS-1.

It has been shown that ATFS-1$^{\Delta 1-32}$, with impaired mitochondrial targeting sequence, could be expressed in the nucleus of HeLa cells (*Nargund et al., 2012*). Therefore, it might be possible to directly test the transcriptional activity of ATFS-1$^{\Delta 1-32}$ in mammalian system. We employed a luciferase reporter assay, in which transcription factor of interest (e.g. wild type ATFS-1$^{\Delta 1-32}$, ATFS-1$^{\Delta 1-32}$ K326R or SMO-1-ATFS-1$^{\Delta 1-32}$) is fused with Gal4 binding domain (BD) to drive the expression of

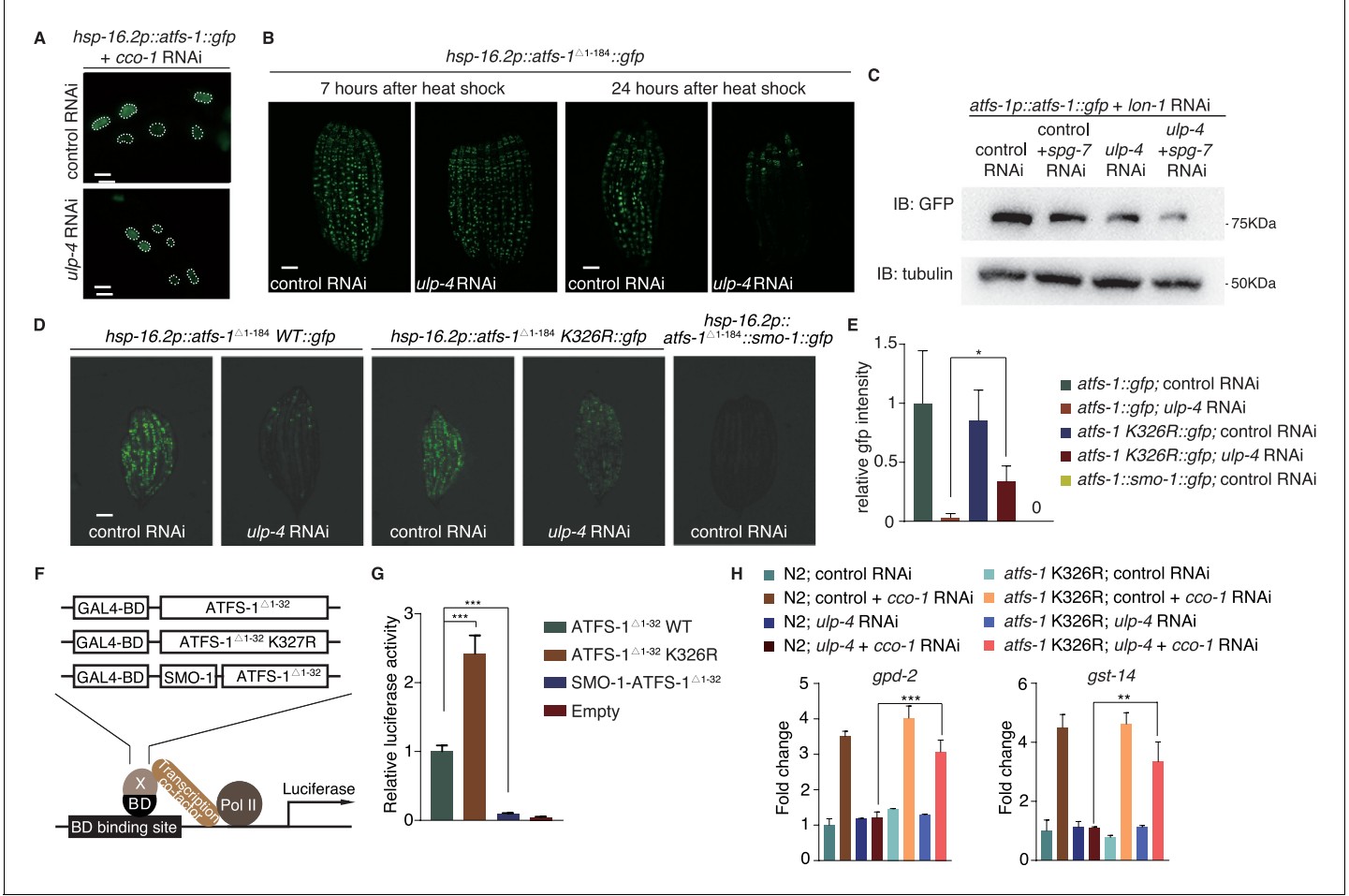

**Figure 5.** SUMOylation affects the stability and transcriptional activity of ATFS-1. (**A**) Representative fluorescent images of ATFS-1::GFP localization with indicated treatment. (**B**) Representative fluorescent images of ATFS-1 $^{\Delta 1-184}$::GFP. Worms were heat-shocked at 37°C for 1 hr and photographed at indicated time points. (**C**) Immunoblot of ATFS-1::GFP expression in *atfs-1p::atfs-1::gfp* worms fed with indicated RNAi. (**D**) Representative fluorescent images of wild-type, SUMOylation mutant or SUMOylation mimetic ATFS-1 $^{\Delta 1-184}$::GFP with indicated RNAi. (**E**) Statistic result of (**D**). (**F**) Diagram of transcriptional activity assay. ATFS-1 $^{\Delta 1-32}$ was cloned into a vector containing Gal4-BD at N-terminus to induce luciferase expression. (**G**) Transcriptional activity assay of ATFS-1 $^{\Delta 1-32}$, ATFS-1 $^{\Delta 1-32}$ K326R or SMO-1-ATFS-1 $^{\Delta 1-32}$. Student's t-test. ***p<0.0002. Error bars indicate standard deviation. (**H**) Quantitative PCR of *god-2* and *gst-14* mRNA levels in wild-type or *atfs-1* K326R worms fed with indicated RNAi. .

DOI: https://doi.org/10.7554/eLife.41792.011

The following figure supplement is available for figure 5:

**Figure supplement 1.** SUMOylation affects the stability and transcriptional activity of ATFS-1.

DOI: https://doi.org/10.7554/eLife.41792.012

luciferase (*Figure 5F* and *Figure 5—figure supplement 1D*). We found that mutation of ATFS-1 SUMOylation site greatly enhanced its transcriptional activity, whereas SUMO-mimetic significantly impaired the activity (*Figure 5G*). Lastly, transcriptions of genes known to be driven by ATFS-1 (eg. *gpd-2* and *gst-14*) (*Nargund et al., 2012*; *Nargund et al., 2015*) were blocked by *ulp-4* RNAi, which could be partially rescued with ATFS-1 K326R mutation (*Figure 5H*). Thus, SUMOylation also impairs the transcriptional activity of ATFS-1.

## *ulp-4* is essential for UPR$^{mt}$-regulated innate immunity and lifespan extension

During mitochondrial stress, UPR$^{mt}$ not only initiates mitochondrial protective responses, but also activates immune responses and extends worm lifespan (*Liu et al., 2014*; *Merkwirth et al., 2016*; *Pellegrino et al., 2014*; *Tian et al., 2016*). The essential function of *ulp-4* in signaling UPR$^{mt}$ makes

it likely to be crucial for animal fitness during mitochondrial stress. Indeed, worms treated with *cco-1* RNAi had a severe synthetic growth defect on *ulp-4* RNAi (**Figure 6A**). Consistently, *ulp-4(tm1597)* mutants revealed a more severe developmental delay when grown on *spg-7* RNAi (**Figure 6—figure supplement 1A–B**). Mutation of the SUMOylation sites of ATFS-1 and DVE-1 in *C. elegans* partially rescued the developmental delay of *spg-7* mutants (**Figure 6—figure supplement 1C–E**). The survival rate of worms exposed to high dosage of antimycin was also significantly reduced in *ulp-4* RNAi (**Figure 6—figure supplement 1F–G**).

A broad range of microbes isolated from natural habitats harboring wild *C. elegans* populations could perturb mitochondrial function and induce the expression of *hsp-6p::gfp* (**Liu et al., 2014**). *ulp-4* RNAi also impaired UPR$^{mt}$ activation when we challenged worms with a *Pseudomonas* strain, a mitochondrial insult isolated from the natural habitat of *C. elegans* (**Liu et al., 2014**) (**Figure 6B**). Moreover, *ulp-4* RNAi suppressed the activation of immune response and xenobiotic detoxification response (**Figure 6C**). To further validate the requirement of *ulp-4* in UPR$^{mt}$-mediated immune response, we treated *irg-1p::gfp* transgenic worms, a reporter strain for pathogen-infected response (**Estes et al., 2010**), with control or *ulp-4* RNAi and then challenged them with *Pseudomonas*. We showed that deficiency of *ulp-4* greatly suppressed the induction of *irg-1* (**Figure 6D**). Deficiency of *ulp-4* also impaired worm development and survival rate when they were infected with *Pseudomonas* (**Figure 6E–F**) (**Kirienko et al., 2014**). The reduced survival rate of *ulp-4*-deficient worms could be rescued with *atfs-1* K326R; *dve-1* K327R mutations, further demonstrating that deSUMOylation of ATFS-1 and DVE-1 is the major function of ULP-4 during mitochondrial stress (**Figure 6G**).

Finally, we analyzed the lifespans of control or *ulp-4* RNAi worms. Under unstressed condition, *ulp-4* RNAi did not affect worm lifespan. However, *ulp-4* RNAi greatly suppressed the lifespan extension in *cco-1* RNAi-treated worms, which could be rescued by mutations of SUMOylation site within ATFS-1 and DVE-1 (**Figure 6H**). Overexpression of *ulp-4* neither affects the basal level of UPR$^{mt}$ under unstressed condition (**Figure 1G**), nor affects worm lifespans with or without mitochondrial stress (**Figure 6—figure supplement 1H**). In contrary, *smo-1* RNAi shortened worm lifespan, but greatly extended the lifespan of *spg-7* mutants (**Figure 6—figure supplement 1I**). Mutations of SUMOylation site within ATFS-1 and DVE-1 extended lifespan of *spg-7* mutants as well (**Figure 6—figure supplement 1J**). Taken together, these results suggest that *ulp-4* is required for mitochondrial stress-induced lifespan extension.

In summary, our studies indicate that mitochondrial stress signals through ULP-4, which deSUMOylates DVE-1 and ATFS-1 to modulate their localization, stability and transcriptional activity. Consequences of these events are elevated innate immunity and prolonged lifespan (**Figure 7**).

## Discussion

We have identified a SUMO-specific peptidase ULP-4 that participates in *C. elegans* UPR$^{mt}$. ULP-4 regulates the entire transcriptional program of UPR$^{mt}$, underscoring the importance of ULP-4-mediated deSUMOylation in UPR$^{mt}$ signaling. However, how mitochondrial stress signals to ULP-4 warrants future analysis.

SUMOylation affects protein function through several mechanisms, including changes of protein conformation, protein–protein interaction, protein stability and subcellular localization (**Chymkowitch et al., 2015**). Interestingly, we find that SUMOylation affects DVE-1 and ATFS-1 through two distinct mechanisms: change of DVE-1 subcellular localization vs. changes of ATFS-1 stability and transcriptional activity. DVE-1 and ATFS-1 constitute the two axes in the transcriptional program of UPR$^{mt}$, each might regulate a different subset of downstream genes. For instance, ATFS-1 has been shown to be the primary factor that controls the expression of genes involved in mitochondrial protein folding, glycolysis, xenobiotic detoxification and immune response (**Nargund et al., 2012**; **Pellegrino et al., 2014**). A detailed analysis of DVE-1 and ATFS-1 substrate selection may facilitate the understanding of why cells employ such intricate regulation of transcriptional response during mitochondrial stress.

DVE-1 is homologous to mammalian SATB class of proteins that function in chromatin remodeling and transcription. Interestingly, it is reported that SATB1 and SATB2 could also be SUMOylated. For example, SUMOylation of SATB2 targets it to the nuclear periphery, where it regulates immunoglobulin μ gene expression (**Dobreva et al., 2003**). SUMOylation of SATB1 targets it to the promyelocytic leukemia (PML) nuclear bodies where it undergoes caspase-mediated cleavage (**Tan et al.,**

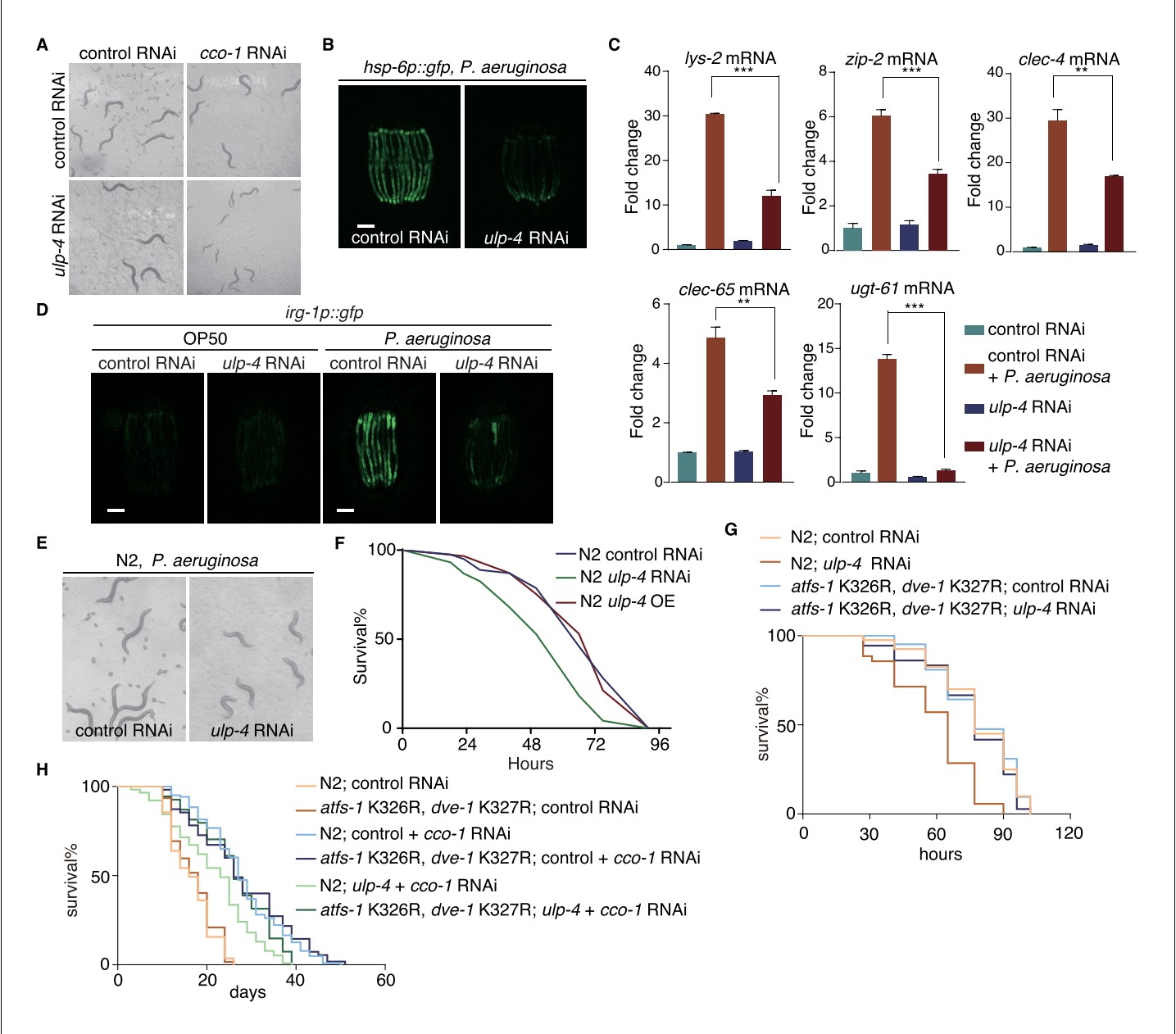

**Figure 6.** *ulp-4* is essential for UPR^mt-regulated innate immunity and lifespan extension. (**A**) Representative images of wild-type worms raised on indicated RNAi. *cco-1* RNAi was treated 16 hr later after control or *ulp-4* RNAi. (**B**) Representative fluorescent images of *hsp-6p::gfp* worms. Animals were pre-treated with control or *ulp-4* RNAi for 24 hr, and then fed with *P. aeruginosa* for additional 48 hr. Student's t-test. Error bars indicate standard deviation. \*\*p<0.002 and \*\*\*p<0.0002. (**C**) Fold changes of immune response gene *lys-2, zip-2, clec-4, clec-65* and *ugt-61* mRNA levels in control or *ulp-4* RNAi animals after exposure to *P. aeruginosa*. (**D**) Immune response reporter strain *irg-1p::gfp* fed with control or *ulp-4* RNAi was infected with *P. aeruginosa* strain PA14. (**E**) Representative images of wild-type worms fed with control or *ulp-4* RNAi for 24 hr and then exposed to *P. aeruginosa*. (**F**) PA14 survival assay in wild-type, *ulp-4* knockdown or overexpression animals. N = 3 biological replicates, n > 40 worms per replicate. Analyzed using Log-Rank method and p<0.05 (N2 control RNAi vs N2 *ulp-4* RNAi). (**G**) PA14 survival assay in N2 or *atfs-1* K326R; *dve-1* K327R animals with indicated RNAi. N = 2 biological replicates, n > 35 worms per replicate. Analyzed using Log-Rank method, p<0.05 (N2 control RNAi vs N2 *ulp-4* RNAi), p>0.05 (*atfs-1* K326R; *dve-1* K327R control RNAi vs *atfs-1* K326R; *dve-1* K327R *ulp-4* RNAi). (**H**) Representative lifespan result of N2 or *atfs-1* K326R; *dve-1* K327R animals fed with indicated RNAi. n > 80 worms per condition. Analyzed using Log-Rank method, p<0.0001 (N2 control +*cco-1* RNAi vs N2 *ulp-4* RNAi + *cco-1* RNAi), p<0.01(N2 *ulp-4* +*cco-1* RNAi vs *atfs-1* K326R; *dve-1* K327R *ulp-4* +*cco-1* RNAi).

DOI: https://doi.org/10.7554/eLife.41792.013

The following figure supplement is available for figure 6:

**Figure supplement 1.** *ulp-4* is essential for UPR^mt-regulated innate immunity and lifespan extension.

*Figure 6 continued on next page*

*Figure 6 continued*

DOI: https://doi.org/10.7554/eLife.41792.014

*2008*). SATB1 has also been shown to form a 'cage'-like distribution and anchors specialized DNA sequences onto its network (*Cai et al., 2003*). Histone H3K9 and H3K14 acetylation mark the binding sites of SATB1, whereas in SATB1 deficient cells, these sites are marked by H3K9 methylation (*Cai et al., 2003*). Similarly, studies in *C. elegans* reported that during mitochondrial perturbation, H3K9 di-methylation globally marks chromatin, leaving portions of chromatin open-up where binding of DVE-1 occurs (*Tian et al., 2016*). All these findings point to the conserved function and regulatory mechanisms of DVE-1 and SATB1. Therefore, it will be interesting in the future to directly test if SATB1 functions as mammalian DVE-1 to signal UPR$^{mt}$. Furthermore, ATF5 has been reported to constitute mammalian homolog of ATFS-1 (*Fiorese et al., 2016*). It will also be interesting to see if SUMOylation can affect the stability and activity of ATF5.

Several mitochondrial quality control processes have evolved to maintain and restore proper mitochondrial function, including mitochondrial unfolded protein response (UPR$^{mt}$), mitochondrial dynamics, and mitophagy (*Andreux et al., 2013*). Cells selectively activate each quality control pathway, depending on the stress level of mitochondria (*Andreux et al., 2013*; *Pellegrino et al., 2013*). Mild mitochondrial inhibition is often associated with the activation of UPR$^{mt}$ to maintain and restore proteostasis. As stress exceeds the protective capacity of UPR$^{mt}$, cells may employ mitochondrial fusion to dilute damaged materials, and activate mitochondrial fission to isolate severely damaged mitochondria for removal through mitophagy. Drp1, the central protein that controls mitochondrial fission, could be SUMOylated (*Prudent et al., 2015*). A RING-finger containing protein MAPL functions as the E3 ligase to promote Drp1 SUMOylation on the mitochondria. SUMOylated Drp1

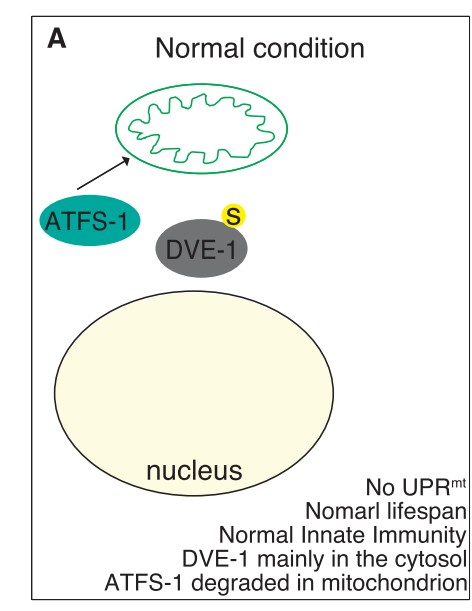
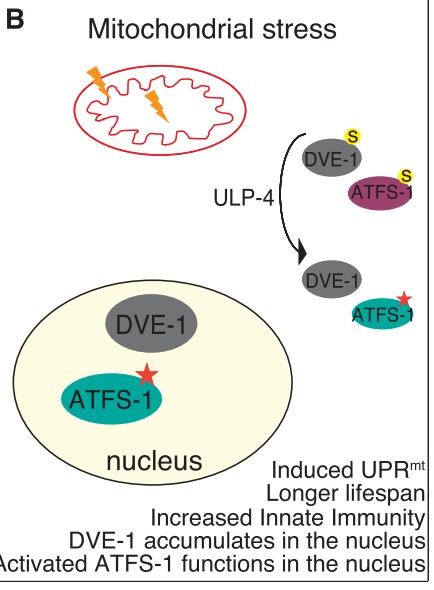
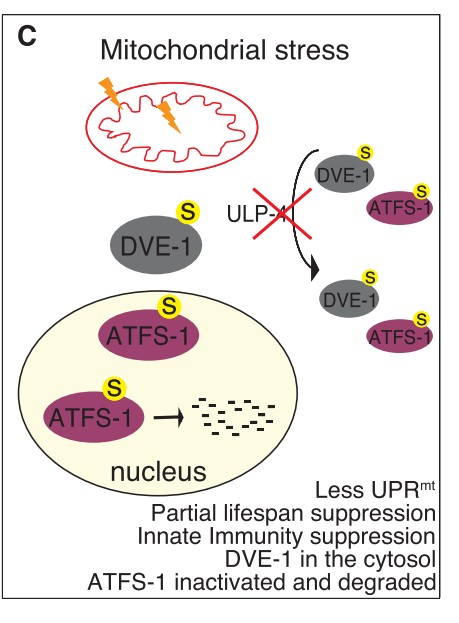

**Figure 7.** Model for ULP-4-mediated UPR$^{mt}$ signaling. (**A**) Under normal condition, ATFS-1 is imported into mitochondria and degraded by mitochondrial protease. DVE-1 is mainly localized in the cytosol. No UPR$^{mt}$ is induced. The animal is normal lived and with normal innate immunity. (**B**) Upon mitochondrial stress, DVE-1 is deSUMOylated by ULP-4 and translocated to the nucleus. Mitochondrial importer efficiency is compromised, leading to ATFS-1 nuclear accumulation. ULP-4 deSUMOylates ATFS-1, leading to increased stability and transcriptional activity (active form). UPR$^{mt}$ is induced. Animals are long-lived, with increased innate immunity. (**C**) Reduction of ULP-4 activity during mitochondrial stress results in SUMOylation of DVE-1 and ATFS-1. SUMOylated of DVE-1 localizes in the cytosol; whereas SUMOylated ATFS-1, which has lower transcriptional activity, is prone to degradation. Induction of UPR$^{mt}$ is impaired. UPR$^{mt}$-regulated innate immunity and lifespan extension is partially suppressed.

DOI: https://doi.org/10.7554/eLife.41792.015

facilitates cristae remodeling, calcium flux and release of cytochrome c, and stabilizes ER/mitochondrial contact sites. Whether SUMOylation affects other proteins in the mitochondrial quality control processes, such as those govern mitophagy, are worth to explore. The discovery of SUMOylation in modulating UPR$^{mt}$ opens up a new research direction to study post-translational regulation of UPR$^{mt}$, and UPR$^{mt}$-mediated immunity and longevity.

# Materials and methods

**Key resources table**

| Reagent type (species) or resource | Designation | Source or reference | Identifiers | Additional information |
|---|---|---|---|---|
| Genetic reagent (*C. elegans*) | hsp-6p::gfp | Caenorhabditis Genetics Center | WB Strain: SJ4100 | |
| Genetic reagent (*C. elegans*) | hsp-16.2p::gfp | Caenorhabditis Genetics Center | WB Strain: CL2070 | |
| Genetic reagent (*C. elegans*) | hsp-4p::gfp | Caenorhabditis Genetics Center | WB Strain: SJ4005 | |
| Genetic reagent (*C. elegans*) | dve-1p::dve-1::gfp | Caenorhabditis Genetics Center | WB Strain: SJ4198 | |
| Genetic reagent (*C. elegans*) | hsp-60p::gfp | Caenorhabditis Genetics Center | WB Strain: SJ4058 | |
| Genetic reagent (*C. elegans*) | N2 | Caenorhabditis Genetics Center | WB Strain: N2 | |
| Genetic reagent (*C. elegans*) | DA2249 (*spg-7* mutant) | Caenorhabditis Genetics Center | WB Strain: DA2249 | |
| Genetic reagent (*C. elegans*) | tm1597 (*ulp-4* mutant) | National Bioresource Project | WB Variation: *tm1597* | |
| Genetic reagent (*C. elegans*) | hsp-60p::gfp; atfs-1 (tm4525) | PMID:22700657 | Dr. Cole Haynes (University of Massachusetts Medical School) | |
| Genetic reagent (*C. elegans*) | hsp-60p::gfp; tm1597 | This paper | N/A | |
| Genetic reagent (*C. elegans*) | dve-1p::dve-1 K327R::gfp | This paper | N/A | |
| Genetic reagent (*C. elegans*) | hsp-16.2p::dve-1::smo-1::gfp | This paper | N/A | |
| Genetic reagent (*C. elegans*) | hsp-16.2p::atfs-1::gfp | This paper | N/A | |
| Genetic reagent (*C. elegans*) | hsp-16.2p::atfs-1Δ1–32::gfp | This paper | N/A | |
| Genetic reagent (*C. elegans*) | hsp-16.2p::atfs-1Δ1–184::gfp | This paper | N/A | |

*Continued on next page*

*Continued*

| Reagent type (species) or resource | Designation | Source or reference | Identifiers | Additional information |
|---|---|---|---|---|
| Genetic reagent (*C. elegans*) | hsp-16.2p::atfs-1Δ1–184::smo-1::gfp | This paper | N/A | |
| Genetic reagent (*C. elegans*) | hsp-16.2p::atfs-1Δ1–184 K326R::gfp | This paper | N/A | |
| Genetic reagent (*C. elegans*) | hsp-60p::gfp;atfs-1(tm4525);atfs-1p::atfs-1 | This paper | N/A | |
| Genetic reagent (*C. elegans*) | hsp-60p::gfp;atfs-1(tm4525);atfs-1p::atfs-1 K326R | This paper | N/A | |
| Genetic reagent (*C. elegans*) | dve-1 K327R | This paper | N/A | Cas9 mutation |
| Genetic reagent (*C. elegans*) | atfs-1 K326R | This paper | N/A | Cas9 mutation |
| Genetic reagent (*C. elegans*) | atfs-1 K326R dve-1 K327R | This paper | N/A | Cas9 mutation |
| Genetic reagent (*C. elegans*) | dve-1 K327R;hsp-6p::gfp | This paper | N/A | Cas9 mutation |
| Genetic reagent (*C. elegans*) | rpl-28p::ulp-4(opti)::gfp | This paper | N/A | |
| Genetic reagent (*C. elegans*) | rpl-28p::ulp-4(opti)::gfp; dve-1p::dve-1::gfp | This paper | N/A | |
| Cell line (*Homo sapiens*) | 293T | ATCC | CRL-3216 | |
| Antibody | Mouse monoclonal anti-GFP | sungen biotech | CAT#KM8009 | (1:1000) |
| Antibody | Rabbit polyclonal anti-GFP | abcam | CAT#ab290 | (1:1000) |
| Antibody | Rabbit polyclonal anti-ATFS-1 | abclonal | N/A | custom made (1:5000) |
| Antibody | Rabbit polyclonal anti-DVE-1 | abclonal | N/A | custom made (1:5000) |
| Antibody | Rabbit polyclonal anti-SMO-1 | abclonal | N/A | custom made (1:5000) |
| Antibody | Mouse monoclonal anti-MYC | CST | CAT#2276 | (1:1000) |
| Antibody | Rat monoclonal anti-TUBULIN | abcam | CAT#6161 | (1:1000) |
| Antibody | Rabbit monoclonal anti-UBC9 | abcam | CAT#ab75854 | (1:1000) |

*Continued on next page*

*Continued*

| Reagent type (species) or resource | Designation | Source or reference | Identifiers | Additional information |
|---|---|---|---|---|
| Strain (*Pseudomonas aeruginosa*) | Pseudomonas aeruginosa | PMID:24695221 | N/A | |
| Strain (*Pseudomonas aeruginosa*) | Pseudomonas aeruginosa | PMID:25274306 | WB Strain: PA14 | |
| Commercial assay or kit | -Trp DO suplement | Coolaber | CAT#PM2140 | |
| Commercial assay or kit | -Leu -Trp DO suplement | Clontech | CAT#630417 | |
| Commercial assay or kit | -Leu -Trp -His DO suplement | Clontech | CAT#630419 | |
| Commercial assay or kit | -Leu -Trp -His -Ade DO suplement | Clontech | CAT#630428 | |
| Commercial assay or kit | -Leu -Trp -His -Met DO suplement | Coolaber | CAT#PM2250 | |
| Commercial assay or kit | Minimal SD Base | Clontech | CAT#630411 | |
| Commercial assay or kit | SYBR Green QPCR mix | Biorad | CAT#172–5122 | |
| Commercial assay or kit | transcript one-step gDNA removal and cDNA synthesis supermix | Transgene | AT311-03 | |
| Commercial assay or kit | Triton X-100 | sigma | T9284 | |
| Commercial assay or kit | Trizol | Cwbio | CW0580A | |
| Commercial assay or kit | Lipofectamine 3000 | life | L3000015 | |
| Commercial assay or kit | Yeast Plasmid Extraction Kit | solarbio | D1160-100 | |
| Commercial assay or kit | dynabeads protei G | life | 10004D | |
| Commercial assay or kit | Protease Inhibitor Cocktail | bimake | B14002 | |
| Commercial assay or kit | Antimycin | sigma | A8674 | |
| Commercial assay or kit | N-Ethylmaleimide | J and K | 128-53-0 | |
| Commercial assay or kit | 5-FLUORO-2'-DEOXYURIDINE | sigma | F0503 | |
| Commercial assay or kit | ECL western blotting kit | Pierce | CAT#32106 | |
| Commercial assay or kit | Dual-Luciferase Reporter Assay Systerm | Promega | CAT#E1910 | |
| Commercial assay or kit | Matchmaker GAL4 Two-Hybrid System 3 | Clontech | CAT# PT3247-1 | |
| Software, algorithm | Graph Pad Prism Software | GraphPad Software | https://www.graphpad.com/scientific-software/prism/ | |
| Software, algorithm | MAFFT | PMID:23329690 | https://www.ebi.ac.uk/Tools/msa/mafft/ | |

*Continued on next page*

*Continued*

| Reagent type (species) or resource | Designation | Source or reference | Identifiers | Additional information |
|---|---|---|---|---|
| Software, algorithm | SMART | PMID: 29040681 | http://smart.embl-heidelberg.de | |
| Software, algorithm | ImageJ | PMID: 22930834 | https://imagej.net/Downloads | |
| Software, algorithm | SUMO-GPS | PMID: 24880689 | http://sumosp.biocuckoo.org | |
| Recombinant DNA reagent | *CMVp-myc-dve-1* | This paper | N/A | |
| Recombinant DNA reagent | *CMVp-myc-dve-1 K327R* | This paper | N/A | |
| Recombinant DNA reagent | *CMVp-myc-33–472 atfs-1* | This paper | N/A | |
| Recombinant DNA reagent | *CMVp-myc-33–472 atfs-1 K326R* | This paper | N/A | |
| Recombinant DNA reagent | *CMVp-gfp-smo-1* | This paper | N/A | |
| Recombinant DNA reagent | *CMVp-ubc9* | This paper | N/A | |
| Recombinant DNA reagent | *CMVp-ulp-4 C48* | This paper | N/A | |
| Recombinant DNA reagent | *adh1p-gal4 BD-dve-1* | This paper | N/A | |
| Recombinant DNA reagent | *adh1p-gal4 BD-dve-1 K327R* | This paper | N/A | |
| Recombinant DNA reagent | *adh1p-gal4 BD-dve-1 K461R* | This paper | N/A | |
| Recombinant DNA reagent | *adh1p-gal4 BD-dve-1 K465R* | This paper | N/A | |
| Recombinant DNA reagent | *adh1p-gal4 BD-dve-1 K327R K355R* | This paper | N/A | |
| Recombinant DNA reagent | *adh1p-gal4 BD-ulp-4* | This paper | N/A | |
| Recombinant DNA reagent | *adh1p-gal4 BD-smo-1* | This paper | N/A | |
| Recombinant DNA reagent | *adh1p-gal4 BD-smo-1 delta GG* | This paper | N/A | |
| Recombinant DNA reagent | *adh1p-gal4 AD-smo-1* | This paper | N/A | |
| Recombinant DNA reagent | *adh1p-gal4 AD-smo-1 delta GG* | This paper | N/A | |
| Recombinant DNA reagent | *adh1p-gal4 AD-smo-1 GG'* | This paper | N/A | |
| Recombinant DNA reagent | *adh1p-gal4 AD-33–184 atfs-1* | This paper | N/A | |
| Recombinant DNA reagent | *adh1p-gal4 AD-185–371 atfs-1* | This paper | N/A | |
| Recombinant DNA reagent | *adh1p-gal4 AD-372–472 atfs-1* | This paper | N/A | |
| Recombinant DNA reagent | *adh1p-gal4 AD-dve-1* | This paper | N/A | |
| Recombinant DNA reagent | *adh1p-gal4 AD-1–150 dve-1* | This paper | N/A | |

*Continued on next page*

*Continued*

| Reagent type (species) or resource | Designation | Source or reference | Identifiers | Additional information |
|---|---|---|---|---|
| Recombinant DNA reagent | *adh1p-gal4 AD-151–300 dve-1* | This paper | N/A | |
| Recombinant DNA reagent | *adh1p-gal4 AD-301–468 dve-1* | This paper | N/A | |
| Recombinant DNA reagent | *adh1p-gal4 AD-ulp-1* | This paper | N/A | |
| Recombinant DNA reagent | *adh1p-gal4 AD-ulp-2* | This paper | N/A | |
| Recombinant DNA reagent | *adh1p-gal4 AD-ulp-4* | This paper | N/A | |
| Recombinant DNA reagent | *adh1p-gal4 AD-ulp-5* | This paper | N/A | |
| Recombinant DNA reagent | *adh1p-gal4 BD-dve-1 met17p-ulp-4* | This paper | N/A | |
| Recombinant DNA reagent | *adh1p-gal4 BD-dve-1 met17p-ulp-5* | This paper | N/A | |
| Recombinant DNA reagent | *adh1p-gal4 BD-dve-1 in pBridge* | This paper | N/A | |
| Recombinant DNA reagent | *Adh1p-gal4 AD-33–184 atfs-1* | This paper | N/A | |
| Recombinant DNA reagent | *Adh1p-gal4 AD-185–371 atfs-1* | This paper | N/A | |
| Recombinant DNA reagent | *Adh1p-gal4 AD-372–472 atfs-1* | This paper | N/A | |
| Recombinant DNA reagent | *CMVp-gal4 BD-33–472 atfs-1* | This paper | N/A | |
| Recombinant DNA reagent | *CMVp-gal4 BD-33–472 atfs-1 K327R* | This paper | N/A | |
| Recombinant DNA reagent | *CMVp-gal4 BD-smo-1-33-472 atfs-1* | This paper | N/A | |
| Recombinant DNA reagent | *rpl-28p::ulp-4(opti)::gfp* | This paper | N/A | |
| Recombinant DNA reagent | *hsp-16.2p::atfs-1::gfp* | This paper | N/A | |
| Recombinant DNA reagent | *hsp-16.2p::33–472 atfs-1::gfp* | This paper | N/A | |
| Recombinant DNA reagent | *hsp-16.2p::185–472 atfs-1::gfp* | This paper | N/A | |
| Recombinant DNA reagent | *hsp-16.2p::185–472 atfs-1 K326R::gfp* | This paper | N/A | |
| Recombinant DNA reagent | *atfs-1p::atfs-1* | This paper | N/A | |
| Recombinant DNA reagent | *atfs-1p::atfs-1 K326R* | This paper | N/A | |
| Recombinant DNA reagent | *dve-1p::dve-1 K327R::gfp* | This paper | N/A | |
| Recombinant DNA reagent | *hsp-16.2p::smo-1::dve-1::gfp* | This paper | N/A | |
| Recombinant DNA reagent | *hsp-16.2p::dve-1::smo-1::gfp* | This paper | N/A | |
| Recombinant DNA reagent | *hsp-16.2p::atfs-1::smo-1::gfp* | This paper | N/A | |

## Worm strains

SJ4100 (zcIs13[*hsp-6p::gfp*]), CL2070 (dvIs70[*hsp-16.2p::gfp*]), SJ4005 (zcIs4[*hsp-4p::gfp*]), SJ4058 (zcIs9[hsp-60p::gfp]), SJ4198 (zcIs39[*dve-1p::dve-1::gfp*]), *ulp-4(tm1597)*, *spg-7(ad2249)* and N2 wild-type worms were obtained from Caenorhabditis Genetics Center. *hsp-60p::gfp;atfs-1*(tm4525) is a generous gift from Dr. Cole Haynes. *dve-1p::dve-1::gfp* plasmid is a gift from Dr. Cole Haynes. We used site-direct mutagenesis to generate *dve-1p::dve-1 K327R::gfp* plasmid and micro-injected into worms.

For generation of *hsp-16.2p::dve-1::smo-1::gfp* worms, *hsp-16.2* promoter, *dve-1::smo-1* and *gfp* sequences were sub-cloned into pDD49.26 vector. Three glycine residues were placed as linker between *smo-1* and *dve-1*.

## Cell line

HEK293T cell was obtained from ATCC, which was authenticated by ATCC. Cells were validated to be free of mycoplasma contamination. No commonly misidentified cell lines were used.

## *C. elegans*, yeast and cell culture

*C. elegans* were cultured at 20°C and fed with *E.coli* OP50 on Nematode Growth Media unless otherwise indicated. Yeast strain AH109 for yeast two-hybrid and three-hybrid assays was cultured with YPDA media at 30°C unless otherwise indicated. 293 T cells were cultured with DMEM (10% fetal bovine serum) at 37°C.

## Induction of UPR$^{mt}$

For RNAi-induced UPR$^{mt}$, RNAi bacteria were grown in LB containing 50 µg/ml carbenicillin at 37°C overnight. 200 µl of RNAi bacteria was seeded onto 6 cm worm plates with 5 mM IPTG. Dried plates were kept at room temperature overnight to allow IPTG induction of dsRNA expression. Synchronized L1 worms were raised on the RNAi plates at 20°C. After 24 hr, 200 µl 10X concentrated RNAi (*atp-2*, *cco-1* or *spg-7*) bacteria were provided. GFP expressions were imaged after 48 hr.

For antimycin A induced UPR$^{mt}$, synchronized L1 worms were raised on 6 cm worm plates for 48 hr. 200 µl of 20 µg/ml antimycin were then provided. Fluorescent images were taken 24 hr after the addition of antimycin.

For *P. aeruginosa* induced UPR$^{mt}$, synchronized L1 worms were raised on 6 cm worm plates for 24 hr before exposure to *P. aeruginosa.* Worms were imaged at adulthood day 1.

## Microscopy

Worms with each indicated fluorescent reporter were dropped in 100 mM NaN3 droplet on 2% agarose pads and imaged with a Zeiss Imager M2 microscopy.

## Western blotting

Worms raised under each described condition were washed off plates with M9 buffer and then washed several times with M9 until supernatant was clear. 2X SDS Laemmli buffer (4% SDS, 20% glycerol, 10% 2-mercaptoethanol, 0.004% bromophenol blue, 0.125M tris-HCl, pH 6.8) was used to re-suspend the worm pellet. Samples were boiled at 95°C for 5 min. Lysates containing the same amount of protein were loaded onto SDS-PAGE and transferred onto PVDF membranes (Bio-Rad). After blocked with 5% non-fat milk, the membrane was probed with the designated first and second antibodies (mouse monoclonal anti-GFP, sungen biotech #KM8009; rabbit polyclonal anti-GFP, abcam #ab290; anti-Myc, CST #2276; anti-tubulin, abcam #ab6161; anti-UBC9, abcam #ab75854; anti-ATFS-1, anti-DVE-1 and anti-SMO-1 antibodies were developed by abclonal), developed with the enhanced chemiluminescence method (Pierce, CAT#32106), and visualized by Tanon 5200 chemical luminescence imaging system. The result analysis was performed by ImageCal (Tanon).

## Immunoprecipitation

293 T cells in 10 cm plate were transfected with 3 µg EGFP-SMO-1, 2 µg UBC9, 5 µg MYC-DVE-1 or MYC-ATFS-1$^{\triangle1-32}$ plasmids via lipofectamine 3000 (Invitrogen, CAT#L3000015) or PEI. For ULP-4 deSUMOylation assay, additional 5 µg MYC-ULP-4 C48 plasmid was co-transfected. 24 hr after transfection, cells were washed with 1X PBS buffer, scraped off the plates and pelleted by centrifugation

at 1000 rpm for 1 min. Immunoprecipitation was performed at 4℃ or on ice. Cell pellet was resus-pended in 1 ml lysis buffer (50 mM tris-HCl pH 7.5, 150 mM NaCl, 1% NP40, 1 mM EDTA, 1 mM EGTA, 20 mM N-Ethylmaleimide, proteinase inhibitor cocktail) and sonicated. Samples were spun down at 21,000 g for 15 min. Supernatants were transferred into new tubes. Protein concentration was quantitated by BCA assay (Thermo, CAT#23225). Same amount of protein wasused for immuno-precipitation with appropriate antibodies. Samples were incubated with agitation at 4℃ for 4 hr. 20 µl pre-washed protein G beads (Invitrogen, CAT#10004D) were added to each sample and rotated for additional 2 hr at 4℃. Protein G beads were washed four times with lysis buffer. 50 ul 2X SDS Laemmli (4% SDS, 20% glycerol, 10% 2-mercaptoenthnol, 0.004 bromophenol blue, 0.125M Tri-HCL, pH 6.8) buffer was added to the beads and boiled for 5 min at 95℃.

## RNA isolation and Q-PCR

Worms were fed with each indicated RNAi or exposed to pathogen. Adulthood day one worms were washed off plates and washed several times with M9 buffer until supernatant was clear. Worm pellets were re-suspended with TRIzol reagent (cwbiol, CAT#cw0580A). Samples were frozen and thawed six times to crack worms. Total RNA was isolated by chloroform extraction, followed by eth-anol precipitation and DNase treatment. cDNA was then synthesized by reverse transcription (trans-gene biotech, CAT#AT311-03). Quantitative real-time PCR was carried out using SYBR GREEN PCR Master Mix (Bio-Rad, CAT#1725121). Quantification of transcripts was normalized to *act-3*.

| Q-PCR primer | Sequence (5' –>3') |
| --- | --- |
| cco-1 F | TCAGTGAAAATAAAACGCGCT |
| cco-1 R | GTTGTTTCCACCTGTTTTGTTCA |
| smo-1 F | AGAGCAGCTGGGCGGATT |
| smo-1 R | CCGAAAACGAAGAGTTTATTTGTAAGATAAATA |
| spg-7 F | CGCCGAACCCGTGATCTATT |
| spg-7 R | GAGTCCTCCGGTACCTGAG |
| ubc-9 F | GTCCATGGGCTGAGTAGTCT |
| ubc-9 R | GGAATACACGGGATTTGTCAACA |
| ulp-1 F | GCAATGGCGATTCGAAATATCC |
| ulp-1 R | TCCAAGATGAACTGGCACCA |
| ulp-2 F | TGGTCAAAGTTCTTCCCGGA |
| ulp-2 R | CACTGCACTATGACGACGTG |
| ulp-4 F | GCTTTCACCCTCTTGCTACA |
| ulp-4 R | AATCAACCGAGGCGCTAGTA |
| ulp-5 F | CCTCATCCTAAGCTCACTCCA |
| ulp-5 R | ACACTTCCAAACGCATCCAA |
| aos-1 F | ATGTGCAATTTCCCACCAGT |
| aos-1 R | ACACCGATATTCACACCAAGA |
| hsp-16.2 F | TCCAGTCTGCAGAATCTCTCC |
| hsp-16.2 R | TGCACCAACATCTACATCTTCAG |
| hsp-4 F | CTCGGAGGAAAGCTCACTGA |
| hsp-4 R | ATGGCTCCTCAGAAGCTTGT |
| atp-2 F | CAAGTCGCTGAGGTGTTCAC |
| atp-2 R | CTTCGGCCTTCTTGAACACA |
| pink-1 F | GTTGCAAAAGGTGGACGACT |
| pink-1 R | AAATGGCCGGAAAACTCGAC |
| hsp-6 F | TCCCAAGTCTTCTCTACCGC |

*Continued on next page*

Continued

| Q-PCR primer | Sequence (5' –>3') |
| --- | --- |
| *hsp-6* R | CACGATCTCTGGCTGAAACG |
| *ugt-61* F | GCAATTGGAGGTCATGACGTAACTATG |
| *ugt-61* R | GCGAAGAATGATTCGGCATCCATCTTG |
| *cyp-14A3* F | CAGTTTCCCGCCGAAAACATCCATTTG |
| *cyp-14A3* R | CAATGCCGTTCTTCTTTGAAGCCTCCAG |
| *act-3* F | TCCCTCGAGAAGTCCTACGA |
| *act-3* R | TCCTGGGTACATGGTGGTTC |
| *clec-4* F | GAGCGACACTGGTGACTGTG |
| *clec-4* R | CCATCCAGAATAGGTTGGCG |
| *lys-2* F | ATCGACTCGAACCAAGCTGCG |
| *lys-2* R | TCGACAGCATTTCCCATTGAAGCGT |
| *zip-2* F | TCGACGAGCAAACGACCTAC |
| *zip-2* R | CTTGTGGCGTGCTCATGTT |
| *clec-65* F | CCCGGTGGTGACTGTGAATA |
| *clec-65* R | AGCTCATATTGTCGCTGGCA |
| *gpd-2* F | TGAAATCCAATGGGGAGCCTC |
| *gpd-2* R | GGAGCAGAGATGATGACCTTCTTG |
| *gst-14* F | GCTACCTTGCTAGAAAATTCGG |
| *gst-14* R | GCCGTTAACTTTTCCAGTTCTT |

## Yeast two-hybrid

Each indicated gene was cloned into pGADT7 or pGBDT7 plasmid (Clontech). Plasmids were then transformed into AH109 yeast strain. We first seeded yeast on –Trp and -Leu double dropout solid culture medium. After yeast colonies were formed, we picked individual yeast colony into sterile water, adjusted it to the same OD and dropped onto -Trp, -Leu, -His and -Ade four dropout solid culture medium. Images of yeast were taken after culturing at 30℃ for 2–4 days.

## Yeast two-hybrid screen

Full-length *dve-1* was cloned into pGBDT7, and transformed into AH109 yeast strain to test for auto-activation and protein expression. We then transformed *C. elegans* AD library plasmids into yeast that already contains *dve-1* BD plasmid. Yeasts were seeded onto four dropout solid culture medium and grown at 30℃ for a week. Each colony was picked, followed by mini-prep to extract AD plasmid for PCR and sequencing.

## Yeast three-hybrid

Yeast three-hybrid for DVE-1 deSUMOylation were carried out using pBRIDGE and pGADT7 vectors. *smo-1* was cloned into pGADT7. *dve-1* alone, or together with *ulp-4/5*, were cloned into pBRIDGE. pGADT7 and pBRIDGE plasmids were transformed into AH109 strain and cultured on -Leu and -Trp double dropout solid culture medium for yeast growth, on -Leu, -Trp and -His dropout medium for SMO-1 and DVE-1 interaction assay, on -Met, -Leu, -Trp and -His dropout medium to induce the expression of ULP-4 or ULP-5 and test for deSUMOylation activity.

## MG132 treatment of worms

Worms were grown on control or *ulp-4* RNAi plates for 48 hr before heat shock at 37℃ for 1 hr. The worms were then transferred in M9 containing indicated RNAi bacteria and 100uM MG132. Images were taken after 24 hr.

## Transcriptional activity assay

Each indicated gene was cloned into pCMV-BD vector. We then transfected 0.5 µg pCMV-BD, 0.5 µg pFR-luci (*Photinus pyralis*) and 0.1 µg pActin-luci (*Renilla reniformis*) into one 24-well of 293 T cells. 24 hr after transfection, we assayed luciferase activity with dual luciferase reporter assay system (Promega, CAT#1910).

## Developmental delay and survival assay

To assay developmental delay induced by *P. aeruginosa*, worms were pre-treated with control or *ulp-4* RNAi at 20℃ for 16 hr. *P. aeruginosa* was then dropped onto worm plates. Images were taken two days later.

To assay survival rate of worms under antimycin treatment, worms were grown at 20℃ on 6-well RNAi plates seeded with control or *ulp-4* RNAi. After 48 hr, 20 µg antimycin A were provided.

## Lifespan analysis

Lifespan analyses were conducted on RNAi plates at 20℃. More than 100 synchronized L1 were seeded onto 6 cm worm plates with control or *ulp-4* RNAi. Animals that did not move when gently touched were scored as dead. Worms were transferred every 2 days to new plates during the first 10 days and were transferred every 3–5 days afterwards. Lifespan experiments were performed twice.

## PA14 survival assay

PA14 survival assay was carried out as described previously (*Kirienko et al., 2014*). PA14 was freshly streaked from frozen stock and cultured 37℃ for 16 hr. 10 µL PA14 were then spread onto 3.5 cm slow killing agar plates (3.5 mg/mL peptone, 3 mg/mL NaCl, 17 mg/mL Agar, 1 mM MgSO4, 25 mM 1MKH2PO4, pH 6, 1 mM CaCla, 5 µg/mL cholesterol). The slow killing plates seeded with PA14 were cultured at 37℃ for 24 hr. FUDR was spread on the plates and L4 worms were picked onto plates and cultured at 25℃. Worms were scored at least three times per day after16 hours, until all worms were dead. PA14 survival assay was performed 2 times with three replications each time.

## CRISPR/Cas9 knock-in

The pDD162 CRISPR/Cas9 expression plasmid was obtained from Addgene (#47549). *C. elegans* Cas9 target prediction tool (https://crispr.cos.uni-heidelberg.de) was used to design target sequences. Templates for recombination were cloned into pDD49.26 vector with ~500 bp overhang upstream and downstream of the target. Cas9 pam sequences were mutated in the templates. The plasmid was injected into worms, with Cas9-*rol-6* as co-injection marker. Rolling worms were singled and validated by PCR and sequencing.

Primers: *dve-1 K327R* F: TTTCCATCAATATTCGACCAGAACCGG

*dve-1 K327R* R: CGAATATTGATGGAAAACAAAGTATCTTGAATAGTTTC.

*atfs-1 K326R* F: TTTTAAGCGTCCAGAAGCATTTTTCCGGGAAGAACCCATG

*atfs-1 K326R* R: CGGAAAAATGCTTCTGGACGCTTAAAAACGTC

## Quantification and statistical analysis

All experiments in this paper, if not specifically indicated, have been repeated for at least three times. Statistical analysis was performed with GraphPad. DVE-1 subcellular localization, transcriptional activity assay, QPCR and worm length were analyzed by Student's t-test. PA14 survival and lifespan assays were analyzed using Log-Rank method. *p<0.05, **p<0.002 and ***p<0.0002.

Worm lengths, relative gfp expression and immunoblot quantification were analyzed by Image J. DVE-1 subcellular localization was counted in more than 40 worms per plate and three independent replicates were analyzed for each condition. Transcriptional activity assay wasanalyzed with three independent replicates for each condition. PA14 survival assay was performed with more than 35 worms per plate and at least two independent replicates were analyzed for each condition. The experiment was repeated for at least two times. Lifespan assay was performed with two biological replicates.

## Acknowledgements

We thank Dr. Cole Haynes for providing *C. elegans* strains and plasmid, and Dr. Xin-Hua Feng for his generous gifts of plasmids. Several *C. elegans* strains used in this work were provided by the National Bio-Resource Project, and the Caenorhabditis Genetics Center, which is supported by the NIH-Officer of Research Infrastructure Programs P40 OD010440. This work is supported by the National Natural Science Foundation of China (grants 91854205 and 31422033), the Ministry of Science and Technology of China (National Key Research and Development Program of China grant 2017YFA0504000973), Peking-Tsinghua Center for Life Sciences, and an HHMI International Research Scholar Award (HHMI#55008739) to YL.

## Additional information

### Funding

| Funder | Grant reference number | Author |
| --- | --- | --- |
| National Natural Science Foundation of China | 91854205 | Ying Liu |
| National Natural Science Foundation of China | 31422033 | Ying Liu |
| Howard Hughes Medical Institute | 55008739 | Ying Liu |
| Ministry of Science and Technology of the People's Republic of China | 2017YFA0504000973 | Ying Liu |

The funders had no role in study design, data collection and interpretation, or the decision to submit the work for publication.

### Author contributions

Kaiyu Gao, Data curation, Formal analysis, Methodology, Writing—original draft; Yi Li, Shumei Hu, Data curation, Formal analysis; Ying Liu, Supervision, Funding acquisition, Investigation, Writing—original draft, Project administration, Writing—review and editing

### Author ORCIDs

Ying Liu https://orcid.org/0000-0002-3328-026X

### Decision letter and Author response

Decision letter https://doi.org/10.7554/eLife.41792.018
Author response https://doi.org/10.7554/eLife.41792.019

## Additional files

### Supplementary files

• Transparent reporting form
DOI: https://doi.org/10.7554/eLife.41792.017

### Data availability

All data generated or analyzed during this study are included in the manuscript and supporting files.

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
