## [Decision Letter]

Thank you for submitting your article "SUMO peptidase ULP-4 regulates mitochondrial UPR-mediated innate immunity and lifespan extension" for consideration by *eLife*. Your article has been reviewed by three peer reviewers, one of whom is a member of our Board of Reviewing Editors, and the evaluation has been overseen by Ivan Dikic as the Senior Editor. The following individuals involved in review of your submission have agreed to reveal their identity: Hui Jiang (Reviewer #2).

The reviewers have discussed the reviews with one another and the Reviewing Editor has drafted this decision to help you prepare a revised submission.

Summary:

In *C. elegans*, mitochondrial stresses trigger mitochondrial unfolded protein response (UPR^mt^) through two transcription factors: DVE-1 and ATFS-1. The signaling cascade from mitochondrial stress to transcription response is incompletely understood. In this manuscript, Gao and colleagues report that SUMO peptidase ULP-4 is critical for UPR^mt^ activation and UPR^mt^-regulated innate immunity and lifespan extension. The authors identify DVE-1 and ATFS-1 as SUMOylated proteins that are deSUMOylated by ULP-4. In the absence of ULP-4, SUMOylated DVE-1 remains in the cytoplasm and SUMOylated ATFS-1 becomes unstable and has reduced transcription activity. Overall, the finding that ULP-4 controls UPR^mt^ in worms is striking and of considerable interest. Particularly, the genetic evidence with ULP-4 mutants exemplifies the major role ULP-4 appears to play in this pathway. This manuscript is suitable for publication in *eLife* after appropriate revisions.

Essential revisions:

1) DVE-1 is SUMOylated at K327 in 293T cells and in worms. ULP-4 deSUMOylates DVE-1 in 293T cells. To fully establish the role of ULP-4 in deSUMOylation of DVE-1 in worms, it is important to examine whether ULP4-RNAi can increase SUMOylation of DVE-1 under normal and stress conditions.

2) While it is convincing that ULP-4 is required for UPR^mt^, the final model (Figure 7) suggests changes in ULP-4 activity/levels upon mitochondrial stress. Otherwise, deSUMOylation of DVE-1 would also appear under normal conditions. Figure 1—figure supplement 1C suggests a small increase in *ulp-4* mRNA. Do ULP-4 levels increase upon induction of UPR^mt^ as potentially suggested in Figure 7? Is overexpression of ULP-4 sufficient to deSUMOylate DVE-1 and drive its transcriptional activity?

3) Findings on the destabilization of ATFS-1 by *ulp-4* RNAi are obtained after overexpression of mislocalized ATFS-1^Δ1-184^ (which lacks the MTS) and studied upon heat shock, which will perturb many processes in addition to mitochondria. Do endogenous ATFS-1 levels (which is usually mitochondrially localized) also change dependent on *ulp-4* RNAi and mitochondrial stress (e.g. *spg-7* RNAi)?

4) Despite the genetic evidence for the role of the K327 or K326 residues of DVE-1 or ATFS-1 respectively for UPR^mt^ induction, actual SUMOylation of these sites remains unclear. The size shifts observed in experiments for both DVE-1 and ATFS-1 are not consistent with mono-SUMOylation of these residues. However, specific deSUMOylation of K327/K326 or two mono-SUMOylations seem unlikely as the authors ruled out the other predicted SUMOylation sites. The authors should explain the observed shifts and confirm SUMOylation on endogenous DVE-1 and ATFS-1. Alternatively, SUMOylation could be confirmed by a mass spec approach showing modification of these sites (similar as for SUMO, see e.g. Impens et al., 2014).

References:

Impens F, Radoshevich L, Cossart P, Ribet D. Mapping of SUMO sites and analysis of SUMOylation changes induced by external stimuli. PNAS. 2014 Aug 26;111(34):12432-7. doi: 10.1073/pnas.1413825111.

---

## [Author Response]

Essential revisions:1) DVE-1 is SUMOylated at K327 in 293T cells and in worms. ULP-4 deSUMOylates DVE-1 in 293T cells. To fully establish the role of ULP-4 in deSUMOylation of DVE-1 in worms, it is important to examine whether ULP4-RNAi can increase SUMOylation of DVE-1 under normal and stress conditions.

We have followed the reviewers’ comments to knockdown *ulp-4* by RNAi and found that *ulp-4* RNAi can increase DVE-1 SUMOylation levels a bit under normal and stress conditions (Figure 2—figure supplement 1D). However, we have to admit that biochemistry on SUMOylation is not easy to do, especially when dealing with worms. Our genetic data is much stronger to support that DVE-1 is the bona fide target of ULP-4 in UPR^mt^ signaling. To further support the function of ULP-4 in deSUMOylation of DVE-1, we overexpressed ULP-4 in worms and observed a reduced level of DVE-1 SUMOylation (Figure 2H). In addition, we have also showed that catalytic domain (C48) of ULP-4 could deSUMOylate DVE-1 when co-transfected in 293T cells (Figure 2I).

2) While it is convincing that ULP-4 is required for UPR^mt^, the final model (Figure 7) suggests changes in ULP-4 activity/levels upon mitochondrial stress. Otherwise, deSUMOylation of DVE-1 would also appear under normal conditions. Figure 1—figure supplement 1C suggests a small increase in ulp-4 mRNA. Do ULP-4 levels increase upon induction of UPR^mt^ as potentially suggested in Figure 7? Is overexpression of ULP-4 sufficient to deSUMOylate DVE-1 and drive its transcriptional activity?

The induction fold of *ulp-4* transcripts is ~1.5, which is comparable to the induction fold of many other mitochondrial stress response genes such as *hsp-6* and *hsp-60* (Figure 1C and Figure 1—figure supplement 1B). We don’t have antibody to detect protein levels of ULP-4. We also tried very hard to generate *ulp-4p::ulp-4::gfp* strain, but haven’t successfully got this strain yet. To further answer the reviewers’ question, we generated *rpl-28p::ulp-4; dve-1p::dve-1::gfp* strain, and showed that overexpression of ULP-4 is sufficient to drive the nuclear accumulation of DVE-1 (Figure 3D). In addition, overexpression of ULP-4 is sufficient to deSUMOylate DVE-1 (Figure 2H). We also tried to use luciferase assay, similar as those in Figure 5F and 5G, to test transcriptional activity of DVE-1. However, we found that expression of DVE-1 alone in 293T cells could not induce target genes transcription. It suggests that DVE-1 may require other co-factors to exert its transcriptional activity.

3) Findings on the destabilization of ATFS-1 by ulp-4 RNAi are obtained after overexpression of mislocalized ATFS-1^Δ1-184^ (which lacks the MTS) and studied upon heat shock, which will perturb many processes in addition to mitochondria. Do endogenous ATFS-1 levels (which is usually mitochondrially localized) also change dependent on ulp-4 RNAi and mitochondrial stress (e.g. spg-7 RNAi)?

We haven’t been able to raise a good ATFS-1 antibody that could detect endogenous ATFS-1 proteins. Therefore, we tried to use an *atfs-1p::atfs-1::gfp* strain and perform western blotting with anti-GFP antibody to detect full-length ATFS-1 proteins. Under normal growth condition, protein level of ATFS-1 is not detectable, because ATFS-1 is imported into mitochondria and degraded by mitochondrial protease LON-1 (Nargund et al., 2012). We fed worms with *lon-1* RNAi and showed that *ulp-4* RNAi reduced protein levels of full-length ATFS-1 (Figure 5C).

4) Despite the genetic evidence for the role of the K327 or K326 residues of DVE-1 or ATFS-1 respectively for UPR^mt^ induction, actual SUMOylation of these sites remains unclear. The size shifts observed in experiments for both DVE-1 and ATFS-1 are not consistent with mono-SUMOylation of these residues. However, specific deSUMOylation of K327/K326 or two mono-SUMOylations seem unlikely as the authors ruled out the other predicted SUMOylation sites. The authors should explain the observed shifts and confirm SUMOylation on endogenous DVE-1 and ATFS-1. Alternatively, SUMOylation could be confirmed by a mass spec approach showing modification of these sites (similar as for SUMO, see e.g. Ingens et al., 2014).

We have followed the reviewers’ suggestion to carry out mass spectrometry analysis. However, we failed to detect SUMOylation on DVE-1 and ATFS-1 through mass spec. Mass spectrometry to detect SUMOylation is not as easy as detecting phosphorylation. A more commonly used approach is to employ site-direct mutagenesis to analyze SUMOylation site within the protein-of-interest.

The size shift might be due to di-SUMOylation on a single lysine residue of the target protein. Similar as ubiquitin, SUMO could also be covalently conjugated and form di- or multi-SUMO chain (Knipsheer et al., 2007; Klug et al., 2013 Molecular Cell; Tatham et al., 2008). To test this idea, we tried to express DVE-1^295-354^ or ATFS-1^261-360^ fragment, which contains only one lysine residue (DVE-1 K327 or ATFS-1 K326) in 293T cells. Only DVE-1^295-354^ expressed well in 293T cells. The size shift of DVE-1^295-354^-3xFLAG upon SMO-1 and UBC9 co-transfection is ~60KDa, similar as the molecular weight of di-EGFP-SUMO (Figure 2—figure supplement 1F). Therefore, we think the size shift that we observed for DVE-1 and ATFS-1 is due to di-SUMO moiety. We have discussed this issue in the revised manuscript.

References:

Klug H, Xaver M, Chaugule VK, Koidl S, Mittler G, Klein F, Pichler A. Ubc9 sumoylation controls SUMO chain formation and meiotic synapsis in *Saccharomyces cerevisiae*. Mol Cell. 2013 Jun 6;50(5):625-36. doi: 10.1016/j.molcel.2013.03.027.

Knipscheer P, van Dijk WJ, Olsen JV, Mann M, Sixma TK. Noncovalent interaction between Ubc9 and SUMO promotes SUMO chain formation. EMBO J. 2007 Jun 6; 26(11): 2797–2807. doi: 10.1038/sj.emboj.7601711

Tatham MH1, Geoffroy MC, Shen L, Plechanovova A, Hattersley N, Jaffray EG, Palvimo JJ, Hay RT. RNF4 is a poly-SUMO-specific E3 ubiquitin ligase required for arsenic-induced PML degradation. Nat Cell Biol. 2008 May;10(5):538-46. doi: 10.1038/ncb1716.